# LATENT DIFFUSION COUNTERFACTUAL EXPLANATIONS

## ABSTRACT

Counterfactual explanations have emerged as a promising method for elucidating the behavior of opaque black-box models. Recently, several works leveraged pixel-space diffusion models for counterfactual generation. To handle noisy, adversarial gradients during counterfactual generation–causing unrealistic artifacts or mere adversarial perturbations–they required either auxiliary adversarially robust models or computationally intensive guidance schemes. However, such requirements limit their applicability, e.g., in scenarios with restricted access to the model's training data. To address these limitations, we introduce Latent Diffusion Counterfactual Explanations (LDCE). LDCE harnesses the capabilities of recent class- or text-conditional foundation latent diffusion models to expedite counterfactual generation and focus on the important, semantic parts of the data. Furthermore, we propose a novel consensus guidance mechanism to filter out noisy, adversarial gradients that are misaligned with the diffusion model's implicit classifier. We demonstrate the versatility of LDCE across a wide spectrum of models trained on diverse datasets with different learning paradigms. Finally, we showcase how LDCE can provide insights into model errors, enhancing our understanding of black-box model behavior.

## 1 INTRODUCTION

Deep learning systems achieve remarkable results across diverse domains (e.g., Brown et al. (2020); Jumper et al. (2021)), yet their opacity presents a pressing challenge: as their usage soars in various applications, it becomes increasingly important to understand their underlying inner workings, behavior, and decision-making processes (Arrieta et al., 2020). There are various lines of work that facilitate a better understanding of model behavior, including: pixel attributions (e.g., Simonyan et al. (2014); Bach et al. (2015); Selvaraju et al. (2017); Lundberg & Lee (2017)), feature visualizations (e.g., Erhan et al. (2009); Simonyan et al. (2014); Olah et al. (2017)), concept-based methods (e.g., Bau et al. (2017); Kim et al. (2018); Koh et al. (2020)), inherently interpretable models (e.g., Brendel & Bethge (2019); Chen et al. (2019); Böhle et al. (2022)), and counterfactual explanations (e.g., Wachter et al. (2017); Goyal et al. (2019)).

In this work, we focus on counterfactual explanations that modify a (f)actual input with the *smallest semantically meaningful* change such that a target model changes its output. Formally, given a (f)actual input $x^F$ and (target) model $f$, Wachter et al. (2017) proposed to find the counterfactual explanation $x^{\mathrm{CF}}$ that achieves a desired output $y^{\mathrm{CF}}$ defined by loss function $\mathcal{L}$ and stays as close as possible to the (f)actual input defined by a distance metric $d$, as follows:

$$x^{\mathrm{CF}} \in \arg\min_{x'} \lambda_c \mathcal{L}(f(x'),\ y^{\mathrm{CF}}) + \lambda_d d(x',\ x^{\mathrm{F}}) \quad . \tag{1}$$

Generating (visual) counterfactual explanations from above optimization problem poses a challenge since, e.g., relying solely on the classifier gradient often results in adversarial examples rather than counterfactual explanations with semantically meaningful (i.e., human comprehensible) changes. Thus, previous works resorted to adversarially robust models (e.g., Santurkar et al. (2019); Boreiko et al. (2022)), restricted the set of image manipulations (e.g., Goyal et al. (2019); Wang et al. (2021)), used generative models (e.g., Samangouei et al. (2018); Lang et al. (2021); Khorram & Fuxin (2022); Jeanneret et al. (2022)), or used mixtures of aforementioned approaches (e.g., Augustin et al. (2022)) to regularize towards the (semantic) data manifold. However, these requirements or restrictions can

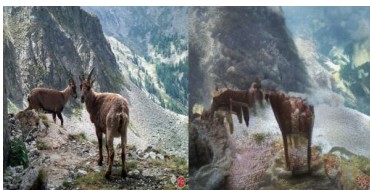

(a) alp → coral reef
on ImageNet with ResNet-50

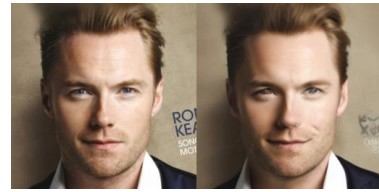

(b) no-smile → smile
on CelebA HQ with DenseNet-121

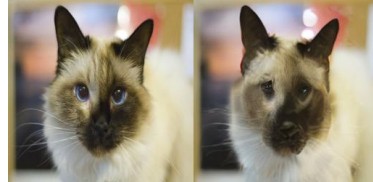

(c) birman → American pit bull
on Oxford Pets with OpenCLIP

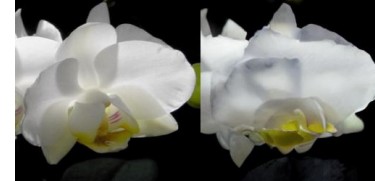

(d) moon orchid → rose
on Oxford Flowers with DINO+linear

Figure 1: LDCE(-txt) can be applied to *any* classifier, is *dataset-agnostic*, and works across various learning paradigms. We show counterfactual explanations (right) for the original image (left) for various dataset and classifier combinations.

hinder the applicability, e.g., in real-world scenarios with restricted data access due to data privacy reasons.

Thus, we introduce Latent Diffusion Counterfactual Explanations (LDCE) which is devoid from such limitations. LDCE leverages recent class- or text-conditional foundational diffusion models combined with a novel *consensus guidance mechanism* that filters out adversarial gradients of the target model that are not aligned with the gradients of the diffusion model's implicit classifier. Moreover, through the decoupling of the semantic from pixel-level details by *latent* diffusion models (Rombach et al., 2022), we not only expedite counterfactual generation but also disentangle semantic from pixel-level changes during counterfactual generation. To the best of our knowledge, LDCE is the first counterfactual approach that can be applied to *any* classifier; independent of the learning paradigm (e.g., supervised or self-supervised) and with extensive domain coverage (as wide as the foundational model's data coverage), while generating high-quality visual counterfactual explanations of the tested classifier; see Figure 1. Code is available at https://anonymous.4open.science/r/ldce.

In summary, our key contributions are the following:

- By leveraging recent class- or text-conditional foundation diffusion models (Rombach et al., 2022), we present the first approach that is both *model-* and *dataset-agnostic* (restricted only by the domain coverage of the foundation model).

- We introduce a novel *consensus guidance mechanism* that eliminates confounding elements, such as an auxiliary classifier, from the counterfactual generation by leveraging foundation models' implicit classifiers as a filter to ensure semantically meaningful changes in the counterfactual explanations of the tested classifiers.

## 2 BACKGROUND

### 2.1 DIFFUSION MODELS

Recent work showed that diffusion models can generate high-quality images (Sohl-Dickstein et al., 2015; Song & Ermon, 2019; Ho et al., 2020; Song et al., 2021; Rombach et al., 2022). The main idea is to gradually add small amounts of Gaussian noise to the data in the forward diffusion process and gradually undoing it in the learned reverse diffusion process. Specifically, given scalar noise scales $\{\alpha_t\}_{t=1}^{T}$ and an initial, clean image $x_0$, the forward diffusion process generates intermediate

noisy representations $\{x_t\}_{t=1}^{T}$, with $T$ denoting the number of time steps. We can compute $x_t$ by

$$x_t = \sqrt{\alpha_t}x_0 + \sqrt{1 - \alpha_t}\epsilon_t, \quad \text{where} \quad \epsilon_t \sim \mathcal{N}(\mathbf{0}, \mathbf{I}) \qquad . \tag{2}$$

The score estimator (i.e., parameterized denoising network) $\epsilon_\theta(x_t, t)$–typically a modified U-Net (Ronneberger et al., 2015)–learns to undo the forward diffusion process for a pair $(x_t, t)$:

$$\epsilon_\theta(x_t, t) \approx \hat{\epsilon}_t = \frac{x_t - \sqrt{\alpha_t}x_0}{\sqrt{1 - \alpha_t}} \qquad . \tag{3}$$

Note that by rewriting Equation 3 (or 2), we can approximately predict the clean data point

$$\hat{x}_0 \approx \frac{x_t - \sqrt{1 - \alpha_t}\epsilon_\theta(x_t, t)}{\sqrt{\alpha_t}} \qquad . \tag{4}$$

To gradually denoise, we can sample the next less noisy representation $x_{t-1}$ with a sampling method $S(x_t, \hat{\epsilon}_t, t) \to x_{t-1}$, such as the DDIM sampler (Song et al., 2021):

$$x_{t-1} = \sqrt{\alpha_{t-1}}\frac{x_t - \sqrt{1 - \alpha_t}\hat{\epsilon}_t}{\sqrt{\alpha_t}} + \sqrt{1 - \alpha_{t-1} - \sigma_t^2}\hat{\epsilon}_t + \sigma_t\epsilon_t \qquad . \tag{5}$$

**Latent diffusion models**     In contrast to GANs (Goodfellow et al., 2020), VAEs (Kingma & Welling, 2014; Rezende et al., 2014), or normalizing flows (Rezende & Mohamed, 2015), (pixel-space) diffusion models' intermediate representations are high-dimensional, rendering the generative process computationally very intensive. To mitigate this, Rombach et al. (2022) proposed to operate diffusion models in a perceptually equivalent, lower-dimensional latent space $\mathcal{Z}$ of a regularized autoencoder $\mathcal{A}(x) = \mathcal{D}(\mathcal{E}(x)) \approx x$ with encoder $\mathcal{E}$ and decoder $\mathcal{D}$ (Esser et al., 2021). Note that this also decouples semantic from perceptual compression s.t. the "focus [of the diffusion model is] on the important, semantic bits of the data" (Rombach et al. (2022), p. 4).

**Controlled image generation**     To condition the generation by some condition $c$, a class label or text, we need to learn a score function $\nabla_x \log p(x|c)$. Through, Bayes' rule we can decompose the score function into an unconditional and conditional component:

$$\nabla_x \log p_\eta(x|c) = \nabla_x \log p(x) + \eta \nabla_x \log p(c|x) \qquad , \tag{6}$$

where the guidance scale $\eta$ governs the influence of the conditioning signal. Note that $\nabla_x \log p(x)$ is just the unconditional score function and $\nabla_x \log p(c|x)$ can be a standard classifier. However, intermediate representations of the diffusion process have high noise levels and directly using a classifier's gradient may result in mere adversarial perturbations (Augustin et al., 2022). To overcome this, previous work used noise-aware classifiers (Dhariwal & Nichol, 2021), optimized intermediate representation of the diffusion process (Jeanneret et al., 2023; Wallace et al., 2023), or used one-step approximations (Avrahami et al., 2022; Augustin et al., 2022; Bansal et al., 2023). In contrast to these works, Ho & Salimans (2022) trained a conditional diffusion model $\nabla_x \log p(x|c)$ with conditioning dropout and leveraged Bayes' rule, i.e.,

$$\nabla_x \log p(c|x) = \nabla_x \log p(x|c) - \nabla_x \log p(x) \qquad , \tag{7}$$

to substitute the conditioning component $\nabla_x \log p(c|x)$ from Equation 6:

$$\nabla_x \log p_\eta(x|c) = \nabla_x \log p(x) + \eta(\nabla_x \log p(x|c) - \nabla_x \log p(x)) \qquad . \tag{8}$$

## 2.2 COUNTERFACTUAL EXPLANATIONS

A counterfactual explanation $x^{\text{CF}}$ is a sample with the *smallest* and *semantically meaningful* change to an original factual input $x^{\text{F}}$ in order to achieve a *desired output*. In contrast to adversarial attacks, counterfactual explanations focus on semantic (i.e., human comprehensible) changes. Initial works on visual counterfactual explanations used gradient-based approaches (Wachter et al., 2017; Santurkar et al., 2019; Boreiko et al., 2022) or restricted the set of image manipulations (Goyal et al., 2019; Akula et al., 2020; Wang et al., 2021; Van Looveren & Klaise, 2021; Vandenhende et al., 2022). Other works leveraged invertible networks (Hvilshøj et al., 2021), deep image priors (Thiagarajan et al., 2021), or used generative models to regularize towards the image manifold to generate high-quality visual counterfactual explanations (Samangouei et al., 2018; Lang et al., 2021; Sauer & Geiger, 2021; Rodriguez et al., 2021; Khorram & Fuxin, 2022; Jacob et al., 2022). Other work generated positive examples through latent transformations for contrastive learning, while enforcing semantic consistency in the latent space of a generative model (Li et al., 2022). Recent works also adopted (pixel-space) diffusion models due to their remarkable generative capabilities (Sanchez & Tsaftaris, 2022; Jeanneret et al., 2022; 2023; Augustin et al., 2022).

---

**Algorithm 1** Latent diffusion counterfactual explanations (LDCE).

---

1: **Input:** (f)actual image $x^{\mathrm{F}}$, condition $c$, target model $f$, encoder $\mathcal{E}$, decoder $\mathcal{D}$, sampler $S$, distance function $d$, "where" function $\phi$, consensus threshold $\gamma$, time steps $T$, weighing factors $\eta$, $\lambda_c$, $\lambda_d$

2: **Output:** counterfactual image $x^{\mathrm{CF}}$

3: $z_T \leftarrow \sqrt{\alpha_t}\mathcal{E}(x^F) + \sqrt{1-\alpha_t}\epsilon_t, \quad \text{where} \quad \epsilon_t \sim \mathcal{N}(\mathbf{0}, \mathbf{I})$      // c.f., Equation 2

4: **for** $t = T, \ldots, 0$ **do**

5:     $\epsilon_{uc}, \ \epsilon_c \leftarrow \epsilon_\theta(z_t, \ t, \ \varnothing), \epsilon_\theta(z_t, \ t, \ c)$

6:     $\hat{x}_0 \leftarrow \mathcal{D}(\frac{z_t - \sqrt{1-\alpha_t}\epsilon_{uc}}{\sqrt{\alpha_t}})$      // c.f., Equation 4

7:     $\texttt{cls\_score}, \ \texttt{dist\_score} \leftarrow \sqrt{1-\alpha_t}\nabla_{z_t}\mathcal{L}(f(\hat{x}_0), \ c), \ \sqrt{1-\alpha_t}\nabla_{z_t}d(\hat{x}_0, x^{\mathrm{F}})$

8:     $\alpha_i \leftarrow \angle(\texttt{cls\_score}_i, (\epsilon_c - \epsilon_{uc})_i)$      // c.f., Equation 10

9:     $\texttt{consensus} \leftarrow \phi(\alpha_i, \ \gamma, \ \texttt{cls\_score}_i, \ \mathbf{0})$      // c.f., Equation 11

10:     $\hat{\epsilon}_t \leftarrow \epsilon_{uc} + \eta \cdot (\lambda_c \frac{\texttt{consensus}}{||\texttt{consensus}||_2} + \lambda_d \frac{\texttt{dist\_score}}{||\texttt{dist\_score}||_2}) \cdot ||\epsilon_c||_2$      // c.f., Equation 6

11:     $z_{t-1} \leftarrow S(z_t, \ \hat{\epsilon}_t, \ t)$      // e.g., Equation 5

12: **end for**

13: $x^{\mathrm{CF}} \leftarrow \mathcal{D}(z_0)$

---

## 3 LATENT DIFFUSION COUNTERFACTUAL EXPLANATIONS (LDCE)

Since generating (visual) counterfactual explanations $x^{\mathrm{CF}}$ is inherently challenging due to high chance of adversarial perturbation by solely relying on the target model's gradient, recent work resorted to (pixel-space) diffusion models to regularize counterfactual generation towards the data manifold by employing the following two-step procedure:

1. **Abduction**: add noise to the (f)actual image $x^{\mathrm{F}}$ through the forward diffusion process, and

2. **Interventional generation**: guide the noisy intermediate representations by the gradients, or a projection thereof, from the target model $f$ s.t. the counterfactual $x^{\mathrm{CF}}$ elicits a desired output $y^{\mathrm{CF}}$ from $f$.

Since intermediate representations of the diffusion model have high-noise levels and may result in mere adversarial perturbations, previous work proposed several schemes to combat these. This included sharing the encoder between target model and denoising network (Diff-SCM, Sanchez & Tsaftaris (2022)), albeit at the cost of model-specificity. Other work (DiME, Jeanneret et al. (2022) & ACE, Jeanneret et al. (2023)) proposed a computationally intensive iterative approach, where at each diffusion time step, they generated an unconditional, clean image to compute gradients on these unnoisy images, but necessitating backpropagation through the entire diffusion process up to the current time step. This results in computational costs of $\mathcal{O}(T^2)$ or $\mathcal{O}(T \cdot I)$ for DiME or ACE, respectively, where $T$ is the number of diffusion steps and $I$ is the number of adversarial attack update steps in ACE. Lastly, Augustin et al. (2022) (DVCE) introduced a guidance scheme involving a projection between the target model and an auxiliary adversarially robust model. However, this requires the auxiliary model to be trained on a very similar data distribution (and task), which may limit its applicability in settings with restricted data access. Further, we found that counterfactuals generated by DVCE are confounded by the auxiliary model; see Appendix A for an extended discussion or Figure 3 of Augustin et al. (2022).

### 3.1 OVERVIEW

We propose Latent Diffusion Counterfactual Explanations (LDCE) that addresses above limitations: LDCE is model-agnostic, computationally efficient, and alleviates the need for an auxiliary classifier requiring (data-specific, adversarial) training. LDCE *harnesses the capabilities of recent class- or text-conditional foundation (latent) diffusion models*, augmented with a novel *consensus guidance mechanism* (Section 3.2). The foundational nature of the text-conditional (latent) diffusion model grants LDCE the versatility to be applied across diverse models, datasets (within reasonable bounds), and learning paradigms, as illustrated in Figure 1. Further, we expedite counterfactual generation by operating diffusion models within a perceptually equivalent, *semantic latent space*, as proposed by Rombach et al. (2022). This also allows LDCE to "focus on the important, semantic [instead of

unimportant, high-frequency details] of the data" (Rombach et al. (2022), p. 4). Moreover, our novel consensus guidance mechanism ensures semantically meaningful changes during the reverse diffusion process by leveraging the implicit classifier (Ho & Salimans, 2022) of class- or text-conditional foundation diffusion models as a filter. Lastly, note that LDCE is compatible to and will benefit from future advancements of diffusion models.

Algorithm 1 provides the implementation outline. We add Gaussian noise to the (f)actual image $x^{\mathrm{F}}$ (line 3) and then guide the backward diffusion process to the counterfactual $x^{\mathrm{CF}}$ s.t. it elicits a desired output $y^{\mathrm{CF}}$ from the target model $f$, while staying close to the (f)actual input (l. 4-13). Below, we describe how we foster semantically meaningful changes by generating counterfactual explanations in the latent space together with our consensus guidance mechanism.

**Counterfactual generation in latent space**     We propose generating counterfactual explanations within a perceptually equivalent, lower-dimensional latent space of an autoencoder (Esser et al., 2021) and rewrite Equation 1 as follows:

$$x^{\mathrm{CF}} = \mathcal{D}(z') \in \underset{z' \in \mathcal{Z} = \{\mathcal{E}(x) | x \in \mathcal{X}\}}{\arg\min} \lambda_c \mathcal{L}(f(\mathcal{D}(z')),\ y^{\mathrm{CF}}) + \lambda_d d(\mathcal{D}(z'),\ x^{\mathrm{F}}) \quad . \tag{9}$$

Yet, we found that generating counterfactual explanations directly in the autoencoder's latent space (Esser et al., 2021) using a gradient-based approach only results in imperceptible, adversarial changes. Conversely, employing a diffusion model on the latent space produced semantically more meaningful changes. Further, it allows the diffusion model to "focus on the important, semantic bits of the data" (Rombach et al. (2022), p. 4) and the autoencoder's decoder $\mathcal{D}$ fills in the unimportant, high-frequency image details. This contrasts prior works that used pixel-space diffusion models for counterfactual generation. We refer to this approach as *LDCE-no consensus*.

## 3.2   CONSENSUS GUIDANCE MECHANISM

To further mitigate the presence of semantically non-meaningful changes, we introduce a novel consensus guidance mechanism (see Figure 2). During the reverse diffusion process, our guidance mechanism exclusively allows for gradients from the target model that align with the freely available implicit classifier of a class- or text-conditional diffusion model (c.f., Equation 7). Consequently, we can use target models out-of-the-box and eliminate the need for auxiliary models that need to be (adversarially) trained on a similar data distribution (and task). We denote these variants as LDCE-cls for class-conditional or LDCE-txt for text-conditional diffusion models, respectively.

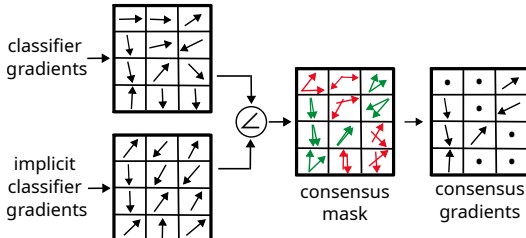

Figure 2:   Our proposed consensus guidance mechanism employs a filtering approach of gradients leveraging the implicit classifier of diffusion models as reference for semantic meaningfulness.

Our consensus guidance mechanism is inspired by the observation that both the gradient of the target model, and the unconditional and conditional score functions of the class- or text-conditional foundation diffusion model (c.f., Equation 7) estimate $\nabla_x \log p(c|x)$. The main idea of our consensus guidance mechanism is to leverage the latter as a reference for semantic meaningfulness to filter out misaligned gradients of the target model that are likely to result in non-meaningful, adversarial modifications. More specifically, we compute the angles $\alpha_i$ between the target model's gradients $\nabla_{z_t} \mathcal{L}(f(\hat{x}_0),\ c)$ and the difference of the conditional and unconditional scores $\epsilon_c - \epsilon_{uc}$ (c.f., Equation 7) for each non-overlapping patch, indexed by $i$:

$$\alpha_i = \angle[(\sqrt{1 - \alpha_t}\nabla_{z_t}\mathcal{L}(f(\hat{x}_0),\ c))_i,\ (\epsilon_c - \epsilon_{uc})_i] \quad . \tag{10}$$

To selectively en- or disable gradients for individual patches, we introduce an angular threshold $\gamma$:

$$\phi_i(\alpha_i,\ \gamma,\ \sqrt{1 - \alpha_t}\nabla_{z_t}\mathcal{L}(f(\hat{x}_0),\ c))_i,\ \mathbf{o}) = \begin{cases} \sqrt{1 - \alpha_t}\nabla_{z_t}\mathcal{L}(f(\hat{x}_0),\ c))_i, & \alpha_i \leq \gamma \\ \mathbf{o}, & \alpha_i > \gamma \end{cases},\tag{11}$$

where $\mathbf{o}$ is the overwrite value (in our case zeros $\mathbf{0}$). Note that by setting the overwrite value $\mathbf{o}$ to zeros, only the target model and the unconditional score estimator–which is needed for regularization towards the data manifold–directly influence counterfactual generation.

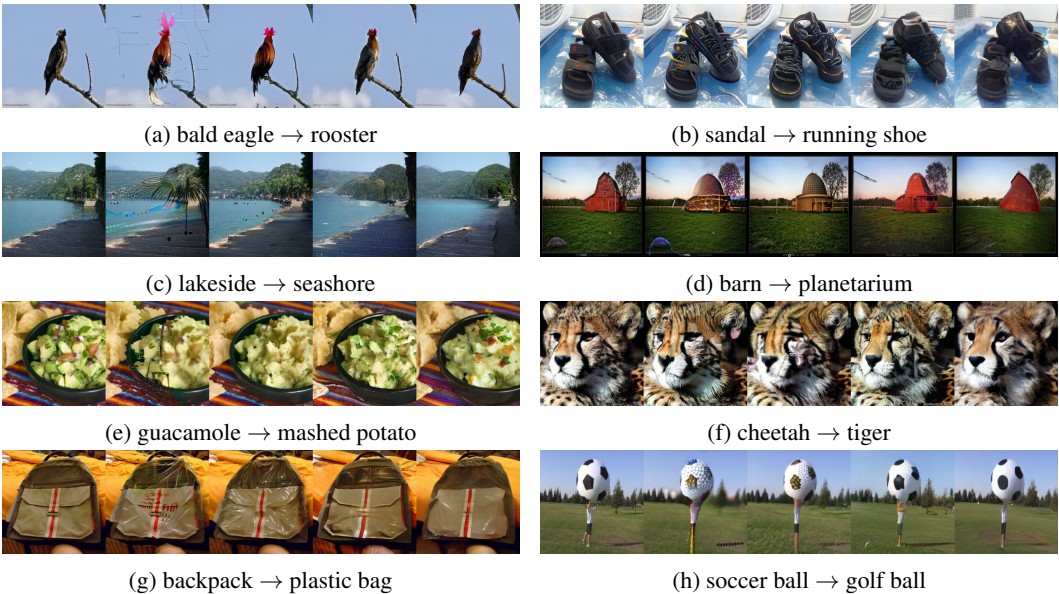

(a) bald eagle → rooster

(b) sandal → running shoe

(c) lakeside → seashore

(d) barn → planetarium

(e) guacamole → mashed potato

(f) cheetah → tiger

(g) backpack → plastic bag

(h) soccer ball → golf ball

Figure 3: Qualitative comparison on ImageNet with ResNet-50. Left to right: original image, counterfactual images for SVCE, DVCE, LDCE-cls, and LDCE-txt. Appendix I provides more examples.

## 4 EXPERIMENTS

**Datasets & models** We evaluated (and compared) LDCE on ImageNet (Deng et al., 2009) (on a subset of 10k images), CelebA HQ (Lee et al., 2020), Oxford Flowers 102 (Nilsback & Zisserman, 2008), and Oxford Pets (Parkhi et al., 2012). All datasets have image resolutions of 256x256. We used ResNet-50 (He et al., 2016), DenseNet-121 (Huang et al., 2017), OpenCLIP-VIT-B/32 (Cherti et al., 2022), and (frozen) DINO-VIT-S/8 with linear classifier (Caron et al., 2021) for ImageNet, CelebA HQ, Oxford Pets or Flowers 102, respectively, as target models. We provide dataset and model licenses in Appendix B and further model details in Appendix C.

**Evaluation protocol** We use two protocols for counterfactual target class selection: **(a) Semantic Hierarchy:** we randomly sample one of the top-4 closest classes based on the shortest path based on WordNet (Miller, 1995). **(b) Representational Similarity:** we compute the instance-wise cosine similarity with SimSiam (Chen & He, 2021) features of the (f)actual images $x^F$ and randomly sample one of the top-5 classes. Note that the latter procedure does not require any domain expertise compared to the former. We adopted the former for ImageNet and the latter for Oxford Pets and Flowers 102. For CelebA HQ, we selected the opposite binary target class.

The evaluation of (visual) counterfactual explanations is inherently challenging: what makes a good counterfactual is arguably very subjective. Despite this, we used various quantitative evaluation criteria covering commonly acknowledged desiderata. **(a) Validity:** We used Flip Ratio (FR), i.e., does the generated counterfactual $x^{CF}$ yield the desired output $y^{CF}$, and COUT (Khorram & Fuxin, 2022) that additionally takes the sparsity of the changes into account. Further, we used the $S^3$ criterion (Jeanneret et al., 2023) that computes cosine similarity between the (f)actual $x^F$ and counterfactual $x^{CF}$. While it has been originally introduced a closeness criterion, we found that $S^3$ correlates strongly with FR, i.e., we found a high Spearman rank correlation of $-0.83$ using the numbers from Table 2. **(b) Closeness:** We used L1 and L2 norms to assess closeness. However, note that $Lp$ norms can be confounded by unimportant, high-level image details. **(c) Realism:** We used FID and sFID (Jeanneret et al., 2023) to assess realism. In contrast to FID, sFID removes the bias caused by the closeness desiderata. Appendix D provides more details.

**Implementation details** We based LDCE-cls on a class-conditional latent diffusion model trained on ImageNet (Rombach et al., 2022) and LDCE-txt on a fine-tuned variant of Stable Diffusion v1.4 for 256x256 images (Pinkey, 2023). Model licenses and links to the weights are provided

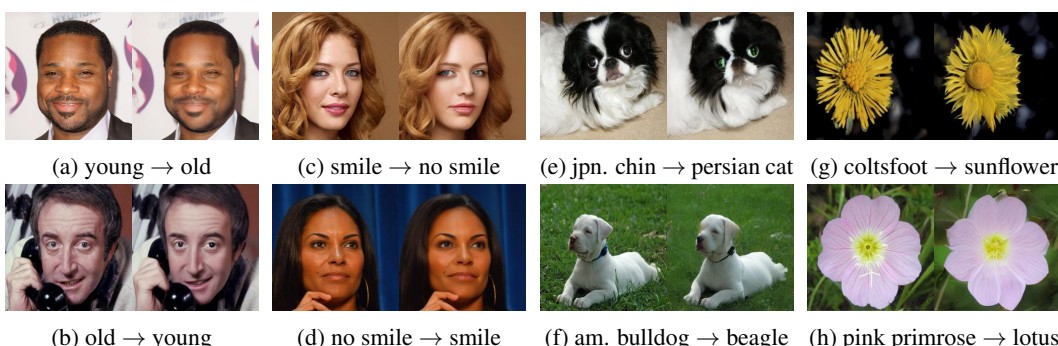

(a) young → old    (c) smile → no smile    (e) jpn. chin → persian cat    (g) coltsfoot → sunflower

(b) old → young    (d) no smile → smile    (f) am. bulldog → beagle    (h) pink primrose → lotus

Figure 4: Qualitative results on CelebA HQ, Oxford Pets, and Flowers 102 using DenseNet-121, CLIP, and DINO with linear classifier, respectively. Left: original image. Right: counterfactual image generated by LDCE-txt. Additional examples are provided in Appendix I.

in Appendix B. For text conditioning, we mapped counterfactual target classes to CLIP-style text prompts (Radford et al., 2021). For our consensus guidance scheme, we used spatial regions of size 1x1 and chose zeros as overwrite values. We used L1 as a distance function $d$ to promote sparse changes. We used a diffusion respacing factor of 2 to expedite counterfactual generation at the cost of a reduction in image quality. We set the weighting factor $\eta$ to 2. We optimized other hyperparameters (diffusion steps $T$ and the other weighing factors $\lambda_c$, $\gamma$, $\lambda_d$) on 10–20 examples. Note that these hyperparameters control the trade-off of the desiderata validity, closeness, and realism. Appendix E provides the selected hyperparameters.

### 4.1 Qualitative evaluation

Figures 1, 3 and 4 show qualitative results for both LDCE-cls and LDCE-txt across a diverse range of models (from convolutional networks to transformers) trained on various real-world datasets (from ImageNet to CelebA-HQ, Oxford-Pets, or Flowers-102) with distinct learning paradigms (from supervision, to vision-only or vision-language self-supervision). We observe that LDCE-txt can introduce local changes, e.g., see Figure 1(b), as well as global modifications, e.g., see Figure 1(a). We observe similar local as well as global changes for LDCE-cls in Figure 3. Notably, LDCE-txt can also introduce intricate changes in the geometry of flower petals without being explicitly trained on such data, see Figure 1(d) or the rightmost column of Figure 4. Further, we found that counterfactual generation gradually evolves from coarse (low-frequency) features (e.g., blobs or shapes) at the earlier time steps towards more intricate (high-frequency) details (e.g., textures) at later time steps. We explores this further in Appendix F. Lastly, both LDCE variants can generate a diverse set of counterfactuals, instead of only single instances, by introducing stochasticity in the abduction step (see Appendix G for examples).

Figure 3 also compares LDCE with previous works: SVCE (Boreiko et al., 2022) and DVCE (Augustin et al., 2022). We found that SVCE often generates high-frequency (Figure 3(c)) or copy-paste-like artifacts (Figure 3(e)). Further, DVCE tends to generate blurry and lower-quality images. This is also reflected in its worse (s)FID scores in Table 1. Moreover, note that counterfactuals generated by DVCE are confounded by its auxiliary model; refer to Appendix A for an extended discussion. In contrast to SVCE and DVCE, both LDCE variants generate fewer artifacts, less blurry, and higher-quality counterfactual explanations. However, we also observed failure modes (e.g., distorted secondary objects) and provide examples in Appendix K. We suspect that some of these limitations are inherited from the underlying foundation diffusion model and, in part, to domain shift. We further discuss these challenges in Section 5.

### 4.2 Quantitative evaluation

We quantitatively compared LDCE to previous work ($\ell_{1.5}$-SVCE (Boreiko et al., 2022), DVCE (Augustin et al., 2022), and ACE (Jeanneret et al., 2023)) on ImageNet. Note that other previous work is hardly applicable to ImageNet or code is not provided, e.g., C3LT (Khorram & Fuxin, 2022). Note that we used a multiple-norm robust ResNet-50 (Croce & Hein, 2021) for $\ell_{1.5}$-SVCE since it is

Table 1: Comparison to SVCE and DVCE on ImageNet using ResNet-50.

| Method | L1 ($\downarrow$) | L2 ($\downarrow$) | FID ($\downarrow$) | sFID ($\downarrow$) | FR ($\uparrow$) |
|---|---|---|---|---|---|
| $\ell_{1.5}$-SVCE[†] (Boreiko et al., 2022) | **5038** | **25** | 22.44 | 28.44 | 83.82 |
| DVCE (Augustin et al., 2022) | *9709* | *38* | 15.1 | 21.14 | 99.56 |
| LDCE-no consensus | 12337 | 41 | 21.70 | *27.10* | **98.4** |
| LDCE-cls | 12375 | 42 | **14.03** | **19.25** | 83.1 |
| LDCE-txt* | 11577 | 41 | *21.0* | 26.5 | *84.4* |

[†]: used an adversarially robust ResNet-50. *: diffusion model not trained on ImageNet.

tailored for adversarially robust models. Further, we limited our comparison to ACE to their smaller evaluation protocol for ImageNet due to its computationally intensive nature.

Table 1 shows that both LDCE variants achieve strong performance for the validity (high FR) and realism (low FID figures) desiderata. Unsurprisingly, we find that SVCE generates counterfactuals that are closer to the (f)actual image (lower L$p$ norms) since it specifically constraints optimization within a $\ell_{1.5}$-ball. We also note that unsurprisingly L$p$ norms are higher for both LDCE variants than for the other methods since L$p$ are confounded by unimportant, high-frequency image details that are not part of our counterfactual optimization, i.e., the decoder just fills in these details. This is corroborated by our qualitative inspection in Section 4.1. Furthermore, we found that both LDCE variants consistently outperform ACE across nearly all evaluation criteria; see Table 2. The only exception is FID, which is unsurprising given that ACE enforces sparse changes, resulting in counterfactuals that remain close to the (f)actual images. This is affirmed by ACE's lower sFID scores, which accounts for this. We provide quantitative comparisons with other methods on CelebA HQ in Appendix H. Despite not being trained specifically for faces, LDCE-txt yielded competitive results to other methods.

Table 2: Comparison of LDCE-cls and LDCE-txt (*diffusion model not trained on ImageNet) to ACE on ImageNet with ResNet-50.

| Method | FID | sFID | $S^3$ | COUT | FR |
|---|---|---|---|---|---|
| **Zebra – Sorrel** | | | | | |
| ACE $\ell_1$ | 84.5 | 122.7 | 0.92 | -0.45 | 47.0 |
| ACE $\ell_2$ | **67.7** | **98.4** | 0.90 | -0.25 | 81.0 |
| LDCE-cls | 84.2 | 107.2 | 0.78 | **-0.06** | **88.0** |
| LDCE-txt* | 82.4 | 107.2 | **0.7113** | -0.2097 | 81.0 |
| **Cheetah – Cougar** | | | | | |
| ACE $\ell_1$ | **70.2** | 100.5 | 0.91 | 0.02 | 77.0 |
| ACE $\ell_2$ | 74.1 | 102.5 | 0.88 | 0.12 | 95.0 |
| LDCE-cls | 71.0 | **91.8** | 0.62 | **0.51** | **100.0** |
| LDCE-txt* | 91.2 | 117.0 | **0.59** | 0.34 | 98.0 |
| **Egyptian Cat – Persian Cat** | | | | | |
| ACE $\ell_1$ | **93.6** | 156.7 | 0.85 | 0.25 | 85.0 |
| ACE $\ell_2$ | 107.3 | 160.4 | 0.78 | 0.34 | 97.0 |
| LDCE-cls | 102.7 | **140.7** | 0.63 | 0.52 | **99.0** |
| LDCE-txt* | 121.7 | 162.4 | **0.61** | **0.56** | **99.0** |

Lastly, we evaluated the computational efficiency of diffusion-based counterfactual methods. Specifically, we compared throughput on a single NVIDIA RTX 3090 GPU with 24 GB memory for 20 batches with maximal batch size. We report that LDCE-txt only required 156 s for four counterfactuals, whereas DVCE needed 210 s for four counterfactuals, and ACE took 184 s for just a single counterfactual. Note that DiME, by design, is even slower than ACE. Above throughput differences translate to substantial speed-ups of 34 % and 371 % compared to DVCE or ACE, respectively.

Above results highlight LDCE as a strong counterfactual generation method. In particular, LDCE-txt achieves on par and often superior performance compared to previous approaches, despite being the only method not requiring any component to be trained on the same data as the target model–a property that, to the best of our knowledge, has not been available in any previous work and also enables applicability in real-world scenarios where data access may be restricted.

### 4.3 IDENTIFICATION AND RESOLUTION OF MODEL ERRORS

Counterfactual explanations should not solely generate high-quality images, like standard image generation, editing or prompt-to-prompt methods (Hertz et al., 2023), but should serve to better understand model behavior. More specifically, we showcase how LDCE-txt can effectively enhance our model understanding of ResNet-50 trained with supervision on ImageNet in the context of misclassifications. To this end, we generated counterfactual explanations of a ResNet-50's misclassi-

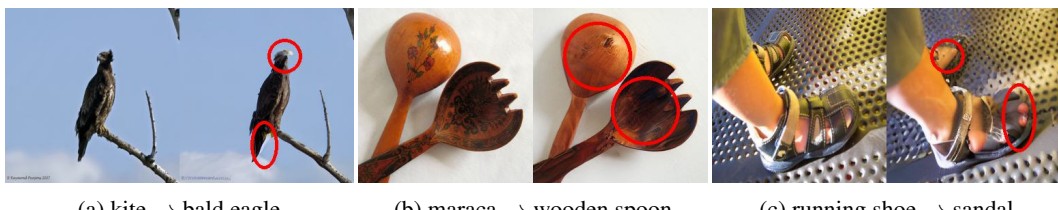

(a) kite → bald eagle          (b) maraca → wooden spoon          (c) running shoe → sandal

Figure 5: Examples of identified classification errors of ResNet-50. Left: original image that is misclassified. Right: counterfactual that is correctly classified. The red ellipses were added manually.

fications towards the true class. As illustrated in Figure 5, this elucidates missing or misleading features in the original image that lead to a misclassification. For instance, it reveals that ResNet-50 may misclassify young bald eagles primarily due to the absence of their distinctive white heads (and tails), which have yet to fully develop. Similarly, we found that painted wooden spoons may be misclassified as maraca, while closed-toe sandals may be confused with running shoes.

To confirm that these findings generalize beyond single instances, we first tried to synthesize images for each error type with InstructPix2Pix (Brooks et al., 2023), but found that it often could not follow the instructions; indicating that these model errors transcend ResNet-50 trained with supervision. Thus, we searched for 50 images on the internet, and report classification error rates of $88\%$, $74\%$, and $48\%$ for the bald eagle, wooden spoon, or sandal model errors, respectively. Finally, we used these images to finetune the last linear layer of ResNet-50. To this end, we separated the 50 images into equally-sized train and test splits; finetuning details are provided in Appendix J. This effectively mitigated these model errors and reduced error rates on the test set by $40\%$, $32\%$, or $16\%$, respectively.

## 5  LIMITATIONS

The main limitation of LDCE is its slow counterfactual generation, which hinders real-time, interactive applications. However, advancements in distilling diffusion models (Salimans & Ho, 2022; Song et al., 2023; Meng et al., 2023) or speed-up techniques (Dao et al., 2022; Bolya et al., 2023) offer promise in mitigating this limitation. Another limitation is the requirement for hyperparameter optimization. Although this is very swift, it still is necessary due to dataset differences and diverse needs in use cases. Lastly, while contemporary foundation diffusion models have expanded their data coverage (Schuhmann et al., 2021), they may not perform as effectively in specialized domains, e.g., for biomedical data, or may contain (social) biases (Bianchi et al., 2023; Luccioni et al., 2023).

## 6  CONCLUSION

We introduced LDCE to generate semantically meaningful counterfactual explanations using class- or text-conditional foundation (latent) diffusion models, combined with a novel consensus guidance mechanism. We show LDCE's versatility across diverse models learned with diverse learning paradigms on diverse datasets, and demonstrate its applicability to better understand model errors and resolve them. Future work could employ our consensus guidance mechanism for counterfactual generation on tabular data or beyond, or LDCE could be extended by incorporating spatial and textual priors.

### BROADER IMPACT

Counterfactual explanations aid in understanding model behavior, can reveal model biases, etc. By incorporating latent diffusion models (Rombach et al., 2022), we make a step forward in reducing computational demands in the generation of counterfactual explanations. However, counterfactual explanations may be manipulated (Slack et al., 2021) or abused. Further, (social) biases in the foundation diffusion models (Bianchi et al., 2023; Luccioni et al., 2023) may also be reflected in counterfactual explanations, resulting in misleading explanations.

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

## A    INFLUENCE OF THE ADVERSARIAL ROBUST MODEL IN DVCE

DVCE (Augustin et al., 2022) uses a projection technique to promote semantically meaningful changes, i.e., they project the unit gradient of an adversarially robust model onto a cone around the unit gradient of the target model, when the gradient directions disagree, i.e., the angle between them exceeds an angular threshold $\alpha$. More specifically, Augustin et al. (2022) defined their cone projection as follows:

$$P_{\text{cone}(\alpha,v)}[w] := \begin{cases} \langle u, w \rangle u, & \angle(w, v) > \alpha \\ v, & \text{else} \end{cases} , \quad (12)$$

where $v, w$ are the unit gradients of the target classifier and adversarially robust classifier, respectively, $\text{cone}(\alpha, v) := \{w \in \mathbb{R}^d : \angle(v, w) \le \alpha\}$, and

$$u = \sin \alpha \frac{P_{v^\perp}(w)}{||P_{v^\perp}(w)||_2} + \cos \alpha \frac{v}{||v||_2} \quad , \quad (13)$$

where $P_{v^\perp}(w) := w - \frac{\langle w, v \rangle}{\langle v, v \rangle} v$. However, due to the

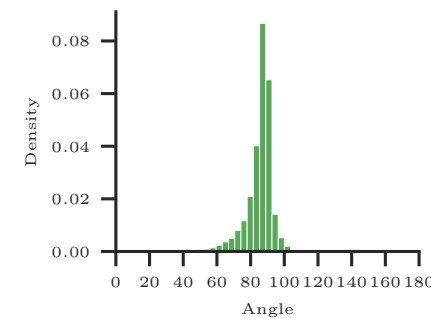

Figure 6: Distribution of angles between the gradient of the target and the robust classifier over 100 images over 100 timesteps.

high dimensionality of the unit gradients ($\mathbb{R}^{256 \cdot 256 \cdot 3}$), they are nearly orthogonal with high probability.[1] In fact, we empirically observed that ca. 97.73 % of the gradients pairs have angles larger than $60°$ during the counterfactual generation of 100 images from various classes; see Figure 6. As a result, we almost always use the cone projection and found that the counterfactuals are substantially influenced by the adversarially robust model: Figure 3 of Augustin et al. (2022) and Figure 7 highlight that the target model has a limited effect in shaping the counterfactuals. Consequently, we *cannot* attribute the changes of counterfactual explanations solely to the target model since they are confounded by the auxiliary adversarially robust model.

## B    DATASET AND MODEL LICENSES

Tables 3 and 4 provide licenses and URLs of the datasets or models, respectively, used in our work. Our implementation is built upon Rombach et al. (2022) (License: Open RAIL-M, URL: https://github.com/CompVis/stable-diffusion) and provided at https://anonymous.4open.science/r/ldce (License: MIT).

Table 3: Licenses and URLs for the datasets used in our experiments.

| Dataset | License | URL |
|---|---|---|
| CelebAMask-HQ (Lee et al., 2020) | CC BY 4.0 | https://github.com/switchablenorms/CelebAMask-HQ |
| Oxford Flowers 102 (Nilsback & Zisserman, 2008) | GNU | https://www.robots.ox.ac.uk/˜vgg/data/flowers/102/ |
| ImageNet (Deng et al., 2009) | Custom | https://www.image-net.org/index.php |
| Oxford Pet (Parkhi et al., 2012) | CC BY-SA 4.0 | https://www.robots.ox.ac.uk/˜vgg/data/pets/ |

Table 4: Licenses and URLs for the target and diffusion models used in our experiments.

| Models | License | URL |
|---|---|---|
| ImageNet class-conditional LDM (Rombach et al., 2022) | MIT | https://github.com/CompVis/latent-diffusion |
| Mini Stable diffusion 1.4 (Pinkey, 2023) | CreativeML Open RAIL-M (Rombach et al., 2022) | https://huggingface.co/justinpinkney/miniSD |
| ResNet-50 for ImageNet (He et al., 2016; TorchVision maintainers and contributors, 2016) | BSD 3 | https://github.com/pytorch/vision |
| Adv. robust ResNet-50 for ImageNet (Boreiko et al., 2022) | MIT | https://github.com/valentyn1boreiko/SVCEs_code |
| DenseNet-121 (Huang et al., 2017) for CelebA HQ (Jacob et al., 2022) | Apache 2 | https://github.com/valeoai/STEEX |
| DINO for Oxford Flowers 102 (Caron et al., 2021) | Apache 2 | https://github.com/facebookresearch/dino |
| OpenCLIP for Oxford Pets (Cherti et al., 2022) | Custom | https://github.com/mlfoundations/open_clip |
| SimSiam (Chen & He, 2021) | CC BY-NC 4.0 | https://github.com/facebookresearch/simsiam |
| CelebA HQ Oracle (Jacob et al., 2022) | Apache 2 | https://github.com/valeoai/STEEX |
| Ported VGGFace2 model from (Cao et al., 2018) | MIT | https://github.com/cydonia999/VGGFace2-pytorch |

---

[1]Note that two randomly uniform unit vectors a nearly orthogonal with high probability in high-dimensional spaces. This can be proven via the law of large numbers and central limit theorem.

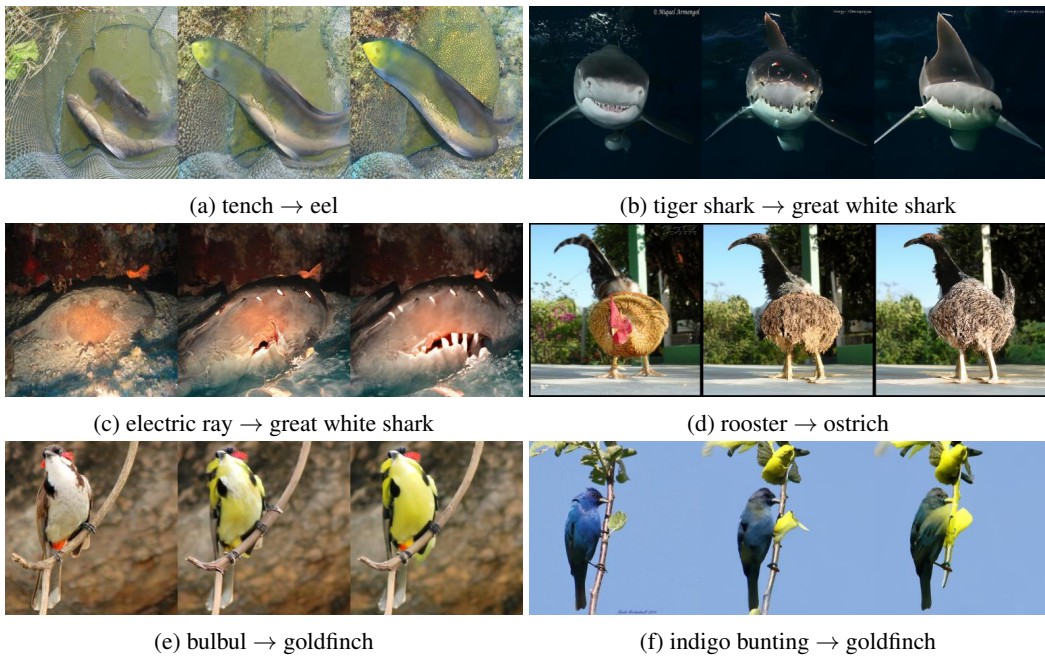

(a) tench → eel

(b) tiger shark → great white shark

(c) electric ray → great white shark

(d) rooster → ostrich

(e) bulbul → goldfinch

(f) indigo bunting → goldfinch

Figure 7: Qualitative examples illustrating the marginal influence of the target model in DVCE. From left to right: original image, counterfactual images generated using DVCE with cone projection using an angular threshold of $30°$ and the robust classifier, and DVCE using the robust classifier only, i.e., without the target model.

## C  MODEL DETAILS

Below, we provide model details:

- **ResNet50 (He et al., 2016) on ImageNet (Deng et al., 2009)**: We used the pretrained ResNet-50 model provided by torchvision (TorchVision maintainers and contributors, 2016).

- **DenseNet-121 (Huang et al., 2017) on CelebA HQ (Lee et al., 2020)**: We used the pretrained DenseNet-121 model provided by Jacob et al. (2022).

- **OpenCLIP (Radford et al., 2021; Cherti et al., 2022) on Oxford Pets (Parkhi et al., 2012)**: We used the provided weights of OpenCLIP ViT-B/32 (Dosovitskiy et al., 2021), and achieved a top-1 zero-shot classification accuracy of $90.5\,\%$ using CLIP-style prompts (Radford et al., 2021).

- **DINO+linear (Caron et al., 2021) on Oxford Flowers 102 (Nilsback & Zisserman, 2008)**: We used the pretrained DINO ViT-S/8 model, added a trainable linear classifier, and trained it on Oxford Flowers 102 for 30 epochs. We used SGD with a learning rate of 0.001 and momentum of 0.9, and cosine annealing (Loshchilov & Hutter, 2019). The model achieved a top-1 classification accuracy of $92.82\,\%$.

## D  EVALUATION CRITERIA FOR COUNTERFACTUAL EXPLANATIONS

In this section, we discuss the evaluation criteria used to quantitatively assess the quality of counterfactual explanations. Even though quantitative assessment of counterfactual explanations is arguably very subjective, these evaluation criteria build a basis of quantitative evaluation based on the commonly recognized desiderata validity, closeness, and realism.

FLIP RATIO (FR)

This criterion focuses on assessing the validity of $N$ counterfactual explanations by quantifying the degree to which the original class label $y_i^{\text{F}}$ of the original image $x_i^{\text{F}}$ flips the target classifier's prediction $f$ to the counterfactual target class $y_i^{\text{CF}}$ for the counterfactual image $x_i^{\text{CF}}$:

$$\text{FR} = \frac{\sum_{i=1}^{N} \mathbb{I}(f(x_i^{\text{CF}}) = y_i^{\text{CF}})}{N} \quad , \tag{14}$$

where $\mathbb{I}$ is the indicator function.

COUNTERFACTUAL TRANSITION (COUT)

Counterfactual Transition (COUT) (Khorram & Fuxin, 2022) measures the sparsity of changes in counterfactual explanations, incorporating validity and sparsity aspects. It quantifies the impact of perturbations introduced to the (f)actual image $x^{\text{F}}$ using a normalized mask $m$ that represents relative changes compared to the counterfactual image $x^{\text{CF}}$, i.e., $m = \delta(||x^{\text{F}} - x^{\text{CF}}||_1)$, where $\delta$ normalizes the absolute difference to $[0, 1]$. We progressively perturb $x^{\text{F}}$ by inserting top-ranked pixel batches from $x^{\text{CF}}$ based on these sorted mask values.

For each perturbation step $t \in \{0, \ldots, T\}$, we record the output scores of the classifier $f$ for the (f)actual class $y^{\text{F}}$ and the counterfactual class $y^{\text{CF}}$ throughout the transition from $x^0 = x^{\text{F}}$ to $x^T = x^{\text{CF}}$. From this, we can compute the COUT score:

$$\text{COUT} = \text{AUPC}(y^{\text{CF}}) - \text{AUPC}(y^{\text{F}}) \in [-1, 1] \quad , \tag{15}$$

where the area under the Perturbation Curve (AUPC) for each class $y \in \{y^{\text{F}}, y^{\text{CF}}\}$ is defined as follows:

$$\text{AUPC}(y) = \frac{1}{T} \sum_{t=0}^{T-1} \frac{1}{2} \left( f_y\left(x^{(t)}\right) + f_y\left(x^{(t+1)}\right) \right) \in [0, 1] \quad . \tag{16}$$

A high COUT score indicates that a counterfactual generation approach finds sparse changes that flip classifiers' output to the counterfactual class.

SIMSIAM SIMILARITY (S3)

This criterion measures the cosine similarity between a counterfactual image $x^{\text{CF}}$ and its corresponding (f)actual image $x^{\text{F}}$ in the feature space of a self-supervised SimSiam model $S$ (Chen & He, 2021):

$$S^3(x^{\text{CF}}, x^{\text{F}}) = \frac{\mathcal{S}(x^{\text{CF}}) \cdot \mathcal{S}(x^{\text{F}})}{\|\mathcal{S}(x^{\text{CF}})\|\|\mathcal{S}(x^{\text{F}})\|} \quad . \tag{17}$$

LP NORMS

$Lp$ norms serve as closeness criteria by quantifying the magnitude of the changes between the counterfactual image $x_i^{\text{CF}}$ and original image $x_i^{\text{F}}$:

$$L_p = \frac{1}{N} \sum_{i=1}^{N} \|d_i\|_p \quad , \tag{18}$$

where $0 < p \leq \infty$ and $C, H, W$ are the number of channels, image height, and image width, respectively, and

$$\|d_i\|_p = \left( \sum_{c=1}^{C} \sum_{h=1}^{H} \sum_{w=1}^{W} |x_{i,c,h,w}^{\text{F}} - x_{i,c,h,w}^{\text{CF}}|^p \right)^{\frac{1}{p}} \quad . \tag{19}$$

Note that $Lp$ norms can be confounded by unimportant, high-frequency image details.

MEAN NUMBER OF ATTRIBUTE CHANGES (MNAC)

Mean Number of Attribute Changes (MNAC) quantifies the average number of attributes modified in the generated counterfactual explanations. It uses an oracle model $O_a$ (i.e., VGGFace2 model (Cao et al., 2018)) which predicts the probability of each attribute $a \in \mathcal{A}$, where $\mathcal{A}$ is the entire attributes space. MNAC is defined as follows:

$$\text{MNAC} = \frac{1}{N} \sum_{i=1}^{N} \sum_{a \in \mathcal{A}} \left[ \mathbb{I} \left( \mathbb{I} \left( O_a(x_i^{\text{CF}}) > \beta \right) \neq \mathbb{I} \left( O_a(x_i^{\text{F}}) > \beta \right) \right) \right] \qquad , \qquad (20)$$

where $\beta$ is a threshold (typically set to 0.5) that determines the presence of attributes. MNAC quantifies the counterfactual method's changes to the query attribute $q$, while remaining independent of other attributes. However, a higher MNAC value can wrongly assign accountability for spurious correlations to the counterfactual approach, when in fact they may be artifacts of the classifier.

CORRELATION DIFFERENCE (CD)

Correlation Difference (CD) (Jeanneret et al., 2022) evaluates the ability of counterfactual methods to identify spurious correlations by comparing the Pearson correlation coefficient $c^{q,a}(x)$, of the relative attribute changes $\delta^q$ and $\delta^a$, before and after applying the counterfactual method. For each attribute $a$, the attribute change $\delta^a$ is computed between pairs of samples $i$ and $j$, as $\delta_{i,j}^a = \hat{y}_i^a - \hat{y}_j^a$, using the predicted probabilities $\hat{y}_i^a$ and $\hat{y}_j^a$ from the oracle model $O$ (i.e., VGGFace2 model (Cao et al., 2018)). The CD for a query attribute $q$ is then computed as:

$$\text{CD}_q = \frac{1}{N} \sum_{i=1}^{N} \sum_{a \in \mathcal{A}} |c^{q,a}(x^{\text{CF}}) - c^{q,a}(x^{\text{F}})| \qquad . \qquad (21)$$

FACE VERIFICATION ACCURACY (FVA) & FACE SIMILARITY (FS)

Face Verification Accuracy (FVA) quantifies whether counterfactual explanations for face attributes maintain identity while modifying the target attribute using the VGGFace2 model (Cao et al., 2018), or not. Alternatively, Jeanneret et al. (2023) proposed Face Similarity (FS) that addresses thresholding issues and the abrupt transitions in classifier decisions in FVA when comparing the (f)actual image $x^{\text{F}}$ and its corresponding counterfactual $x^{\text{CF}}$. FS directly measures the cosine similarity between the feature encodings, providing a more continuous assessment (similar to $\text{S}^3$).

FRÉCHET INCEPTION DISTANCES (FID & sFID)

Fréchet Inception Distance (FID) (Heusel et al., 2017) and split FID (sFID) (Jeanneret et al., 2023) evaluate the realism of generated counterfactual images by measuring the distance on the dataset level between the InceptionV3 (Szegedy et al., 2016) feature distributions of real and generated images:

$$\text{FID} = \|\mu_{\text{F}} - \mu_{\text{CF}}\|_2^2 + \text{Tr}(\Sigma_{\text{F}} + \Sigma_{\text{CF}} - 2\sqrt{\Sigma_{\text{F}}\Sigma_{\text{CF}}}) \qquad , \qquad (22)$$

where $\mu_{\text{F}}, \mu_{\text{CF}}$ and $\Sigma_{\text{F}}, \Sigma_{\text{CF}}$ are the feature-wise mean or covariance matrices of the InceptionV3 feature distributions of real and generated images, respectively. However, there is a strong bias in FID towards counterfactual approaches that barely alter the pixels of the (f)actual inputs. To address this, Jeanneret et al. (2023) proposed to split the dataset into two subsets: generate counterfactuals for one subset, compute FID between those counterfactuals and the (f)actual inputs of the untouched subset, and vice versa, and then take the mean of the resulting FIDs.

## E   HYPERPARAMETERS

Table 5 provides our manually tuned hyperparameters. For our LDCE-txt, we transform the counterfactual target classes $y^{\text{CF}}$ to CLIP-style text prompts (Radford et al., 2021), as follows:

- ImageNet: `a photo of a {category name}.`

Table 5: Manually-tuned hyperparameters.

| Hyperparameters | LDCE-cls ImageNet | LDCE-txt ImageNet | CelebA HQ | Flowers | Oxford Pets |
|---|---|---|---|---|---|
| consensus threshold $\gamma$ | 45° | 50° | 55° | 45° | 45° |
| starts timestep $T$ | 191 | 191 | 200 | 250 | 191 |
| classifier weighting $\lambda_c$ | 2.3 | 3.95 | 4.0 | 3.4 | 4.2 |
| distance weighting $\lambda_d$ | 0.3 | 1.2 | 3.3 | 1.2 | 2.4 |

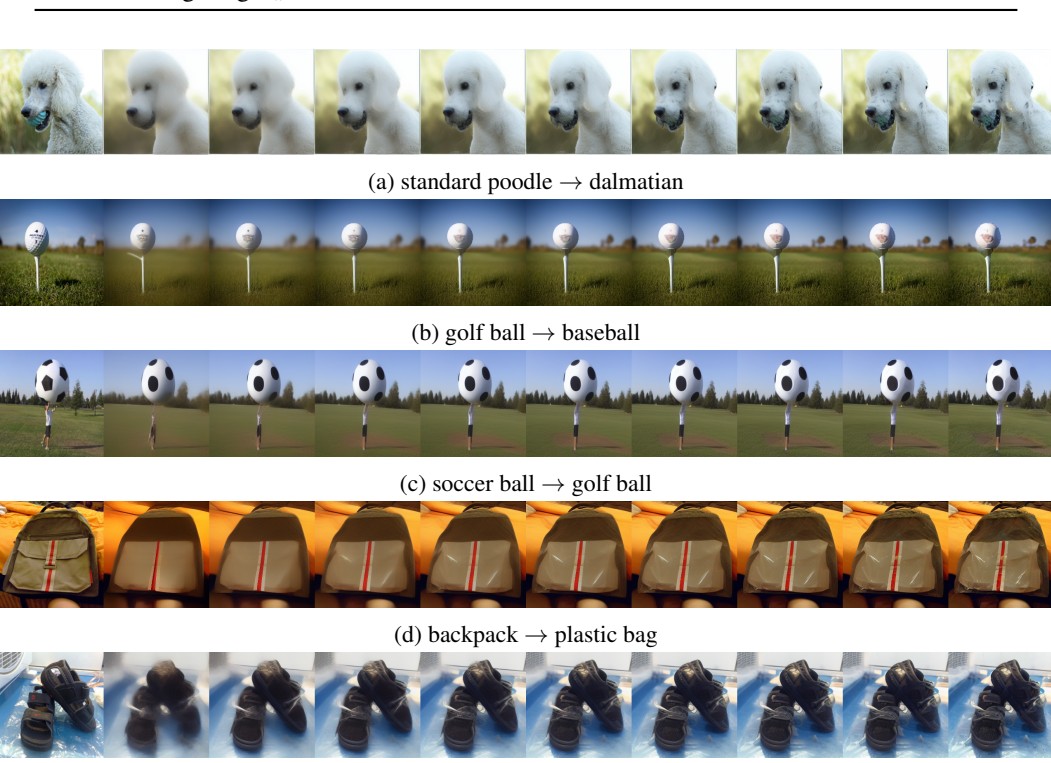

(a) standard poodle → dalmatian

(b) golf ball → baseball

(c) soccer ball → golf ball

(d) backpack → plastic bag

(e) sandal → running shoe

Figure 8: Visualization of changes over the course of the counterfactual generation. From left to right: (f)actual image $x^{\mathrm{F}}$, intermediate visualizations (linearly spaced) till final counterfactual $x^{\mathrm{CF}}$.

- CelebA HQ: `a photo of a {attribute name} person.` (`attribute name` ∈ {non-smiling, smiling, old, young}).
- Oxford Flowers 102: `a photo of a {category name}, a type of flower.`
- Oxford Pets: `a photo of a {category name}, a type of pet.`

We note that more engineered prompts may yield better counterfactual explanations, but we leave such studies for future work.

## F   CHANGES OVER THE COURSE OF THE COUNTERFACTUAL GENERATION

We conducted a deeper analysis to understand how a (f)actual image $x^{\mathrm{F}}$ is transformed into a counterfactual explanation $x^{\mathrm{CF}}$. To this end, we visualized intermediate steps (linearly spaced) of the diffusion process in Figure 8. We found that the image gradually evolves from $x^{\mathrm{F}}$ to $x^{\mathrm{CF}}$ by modifying coarse (low-frequency) features (e.g., blobs or shapes) in the earlier steps and more intricate (high-frequency) features (e.g., textures) in the latter steps of the diffusion process.

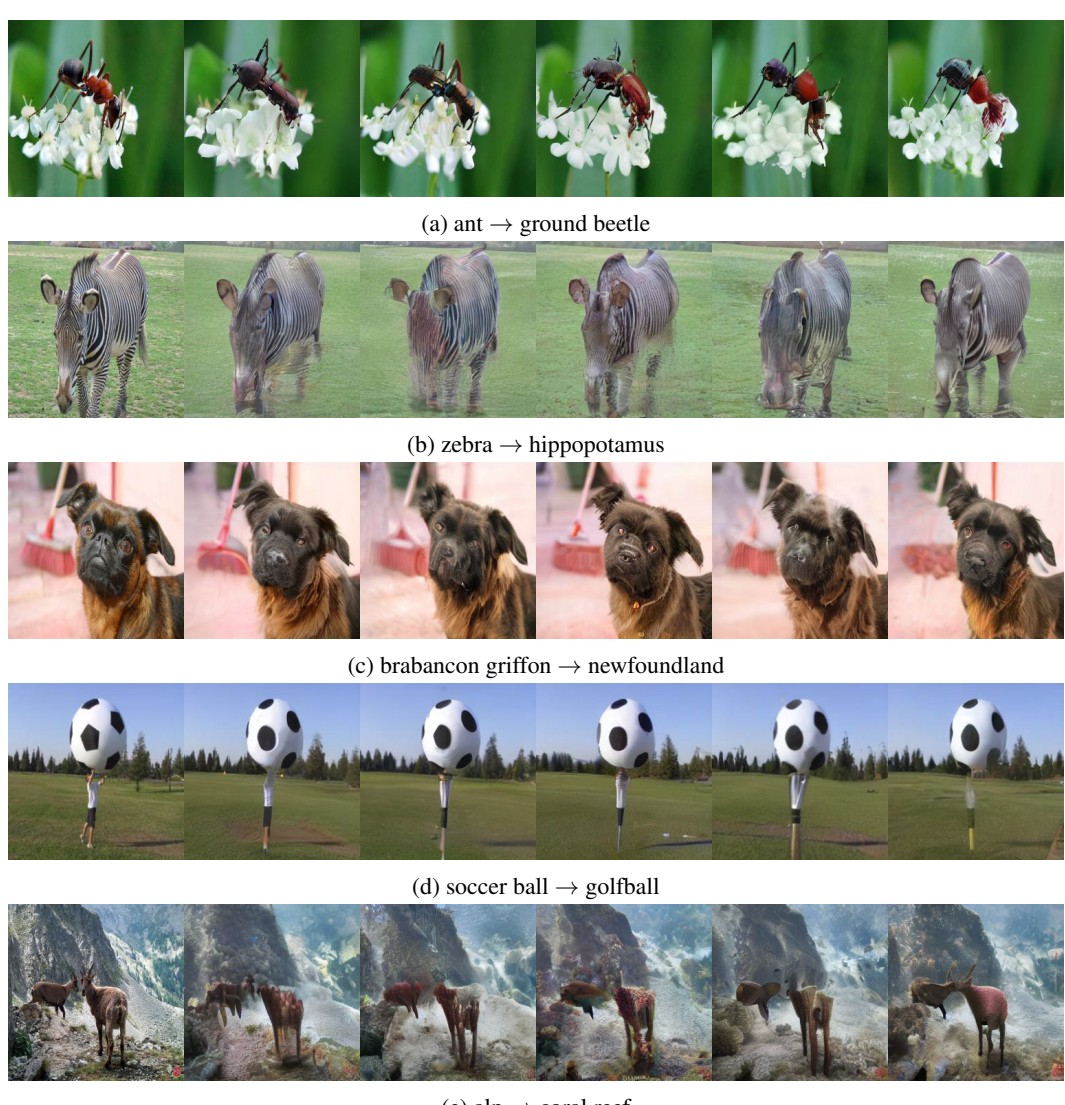

(a) ant → ground beetle

(b) zebra → hippopotamus

(c) brabancon griffon → newfoundland

(d) soccer ball → golfball

(e) alp → coral reef

Figure 9: Qualitative diversity assessment across five different random seeds (0-4) using LDCE-txt on ImageNet (Deng et al., 2009) with ResNet-50 (He et al., 2016). From left to right: original image, counterfactual image generated by LDCE-txt for five different seeds.

## G   DIVERSITY OF THE GENERATED COUNTERFACTUAL EXPLANATIONS

Diffusion models by design are capable of generating image distributions. While the used DDIM sampler (Song et al., 2021) is deterministic, we remark that the abduction step (application of forward diffusion onto the (f)actual input $x^{\mathrm{F}}$) still introduces stochasticity in our approach, resulting in the generation of diverse counterfactual images. More specifically, Figure 9 shows that the injected noise influences the features that are added to or removed from the (f)actual image at different scales. Therefore, to gain a more comprehensive understanding of the underlying semantics driving the transitions in classifiers' decisions, we recommend to generate counterfactuals for multiple random seeds.

## H   QUANTITATIVE RESULTS ON CELEBA HQ

Table 6 provides quantitative comparison to previous methods on CelebA HQ (Lee et al., 2020). We compared to DiVE (Rodriguez et al., 2021), STEEX (Jacob et al., 2022), DiME (Jeanneret et al.,

Table 6: Quantitative comparison to previous works using DenseNet-121 on CelebA HQ. All methods except for ours require access to the target classifier's training data distribution.

| Method | FID ($\downarrow$) | sFID ($\downarrow$) | FVA ($\uparrow$) | FS ($\uparrow$) | MNAC ($\downarrow$) | CD ($\downarrow$) | COUT ($\uparrow$) |
|---|---|---|---|---|---|---|---|
| **Smile** | | | | | | | |
| DiVE (Rodriguez et al., 2021) | 107.0 | - | 35.7 | - | 7.41 | - | - |
| STEEX (Jacob et al., 2022) | 21.9 | - | 97.6 | - | 5.27 | - | - |
| DiME (Jeanneret et al., 2022) | 18.1 | 27.7 | 96.7 | 0.6729 | 2.63 | 1.82 | **0.6495** |
| ACE $\ell_1$ (Jeanneret et al., 2023) | **3.21** | **20.2** | **100.0** | **0.8941** | **1.56** | 2.61 | 0.5496 |
| ACE $\ell_2$ (Jeanneret et al., 2023) | 6.93 | 22.0 | **100.0** | 0.8440 | 1.87 | 2.21 | 0.5946 |
| LDCE-txt | 13.6 | 25.8 | 99.1 | 0.756 | 2.44 | **1.68** | 0.3428 |
| LDCE-txt* | 15.3 | 26.7 | 99.8 | 0.8182 | 2.5 | 2.15 | 0.5938 |
| **Age** | | | | | | | |
| DiVE (Rodriguez et al., 2021) | 107.5 | - | 32.3 | - | 6.76 | - | - |
| STEEX (Jacob et al., 2022) | 26.8 | - | 96.0 | - | 5.63 | - | - |
| DiME (Jeanneret et al., 2022) | 18.7 | 27.8 | 95.0 | 0.6597 | 2.10 | 4.29 | **0.5615** |
| ACE $\ell_1$ (Jeanneret et al., 2023) | **5.31** | **21.7** | **99.6** | **0.8085** | **1.53** | 5.4 | 0.3984 |
| ACE $\ell_2$ (Jeanneret et al., 2023) | 16.4 | 28.2 | **99.6** | 0.7743 | 1.92 | 4.21 | 0.5303 |
| LDCE-txt | 14.2 | 25.6 | 98.0 | 0.7319 | 2.12 | **4.02** | 0.3297 |
| LDCE-txt* | 15.5 | 27.1 | 99.9 | 0.815 | 1.86 | 4.14 | 0.4155 |

*: overwrites values with classifier-free guidance score instead of zeros in our consensus guidance mechanism.

2022), and ACE (Jeanneret et al., 2023) on CelebA HQ (Lee et al., 2020) using a DenseNet-121 (Jacob et al., 2022). Note that in contrast to previous works, LDCE-txt is not specifically trained on a face image distribution and still yields competitive quantitative results. Table 6, Figures 4(a) to 4(d) as well as Figure 11 demonstrate the ability of LDCE-txt to capture and manipulate distinctive facial features, also showcasing its efficacy in the domain of human faces: LDCE-txt inserts or removes local features such as wrinkles, dimples, and eye bags when moving along the smile and age attributes.

## I  ADDITIONAL QUALITATIVE EXAMPLES

Figures 10 to 13 provide additional qualitative examples for ImageNet with ResNet-50, CelebA HQ with DenseNet-121, Oxford Pets with OpenCLIP VIT-B/32, or Oxford Flowers 102 with (frozen) DINO-VIT-S/8 with (trained) linear classifier, respectively. Note that, in contrast to standard image generation, editing or prompt-to-prompt tuning, we are interested in *minimal* semantically meaningful changes to *flip* a target classifier's prediction (and not just generating the best looking image).

## J  FINETUNING DETAILS

We finetuned the final linear layer of ResNet-50 on the ImageNet training set combined with 25 examples that correspond to the respective model error type for 16 epochs and a batch size of 512. We use stochastic gradient descent with learning rate of 0.1, momentum of 0.9, and weight decay of 0.0005. We used cosine annealing as learning rate scheduler and standard image augmentations (random crop, horizontal flip, and normalization). We evaluated the final model on the holdout test set.

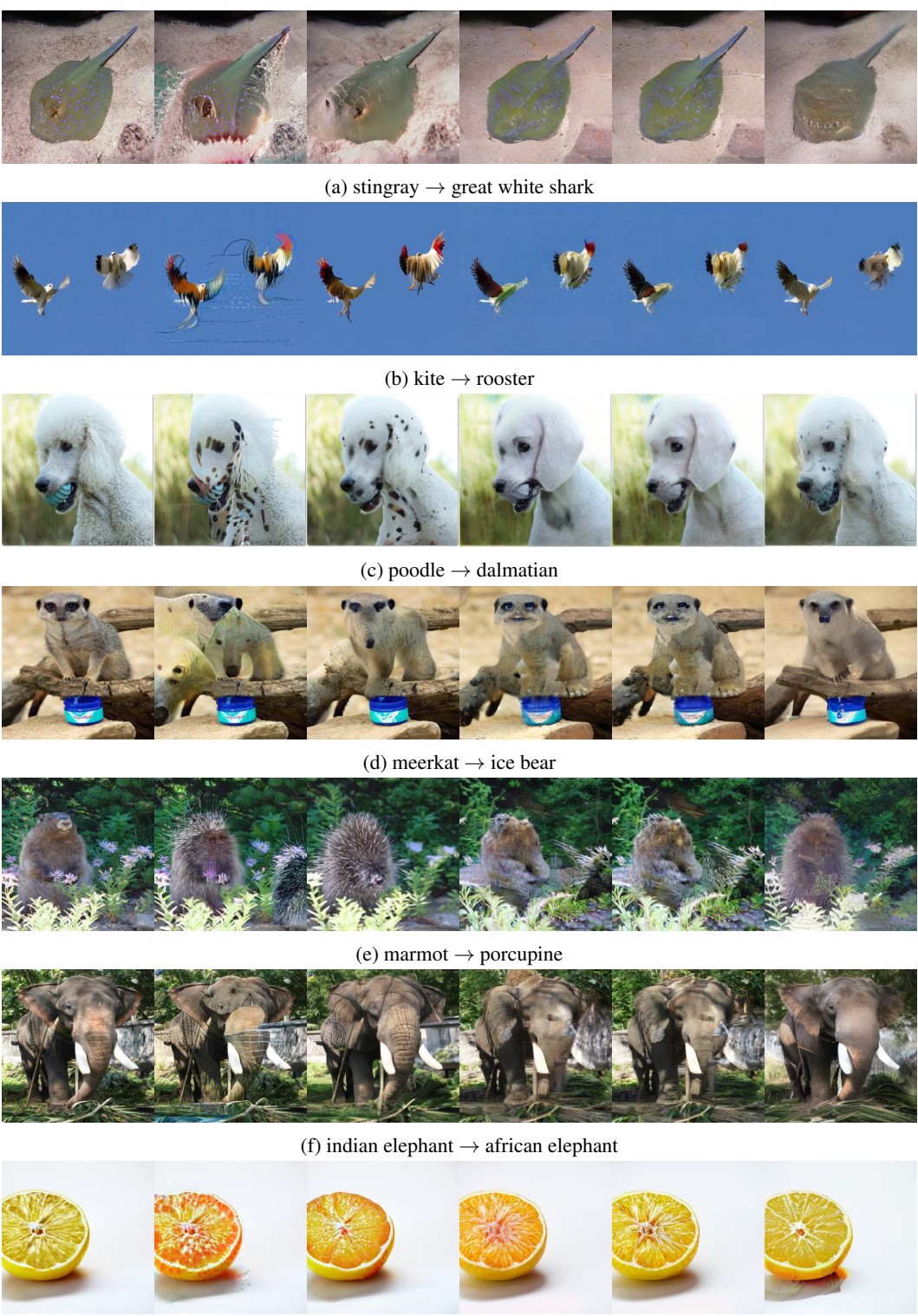

(a) stingray → great white shark

(b) kite → rooster

(c) poodle → dalmatian

(d) meerkat → ice bear

(e) marmot → porcupine

(f) indian elephant → african elephant

(g) lemon → orange

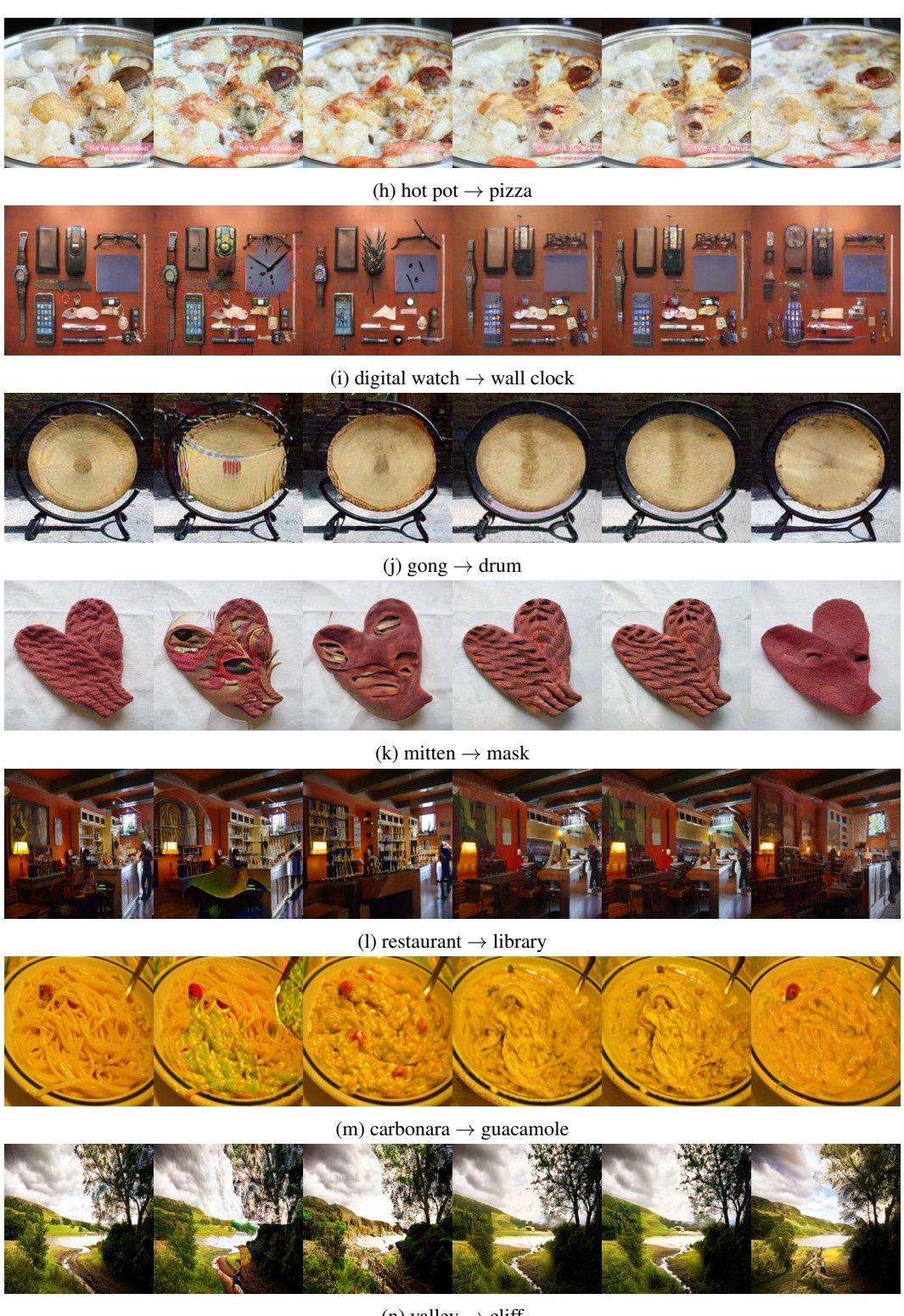

(h) hot pot → pizza

(i) digital watch → wall clock

(j) gong → drum

(k) mitten → mask

(l) restaurant → library

(m) carbonara → guacamole

(n) valley → cliff

Figure 10: Additional qualitative results for on ImageNet with ResNet-50. From left to right: original image, counterfactual images generated by SVCE (Boreiko et al., 2022), DVCE (Augustin et al., 2022), LDCE-no consensus, LDCE-cls, and LDCE-txt.

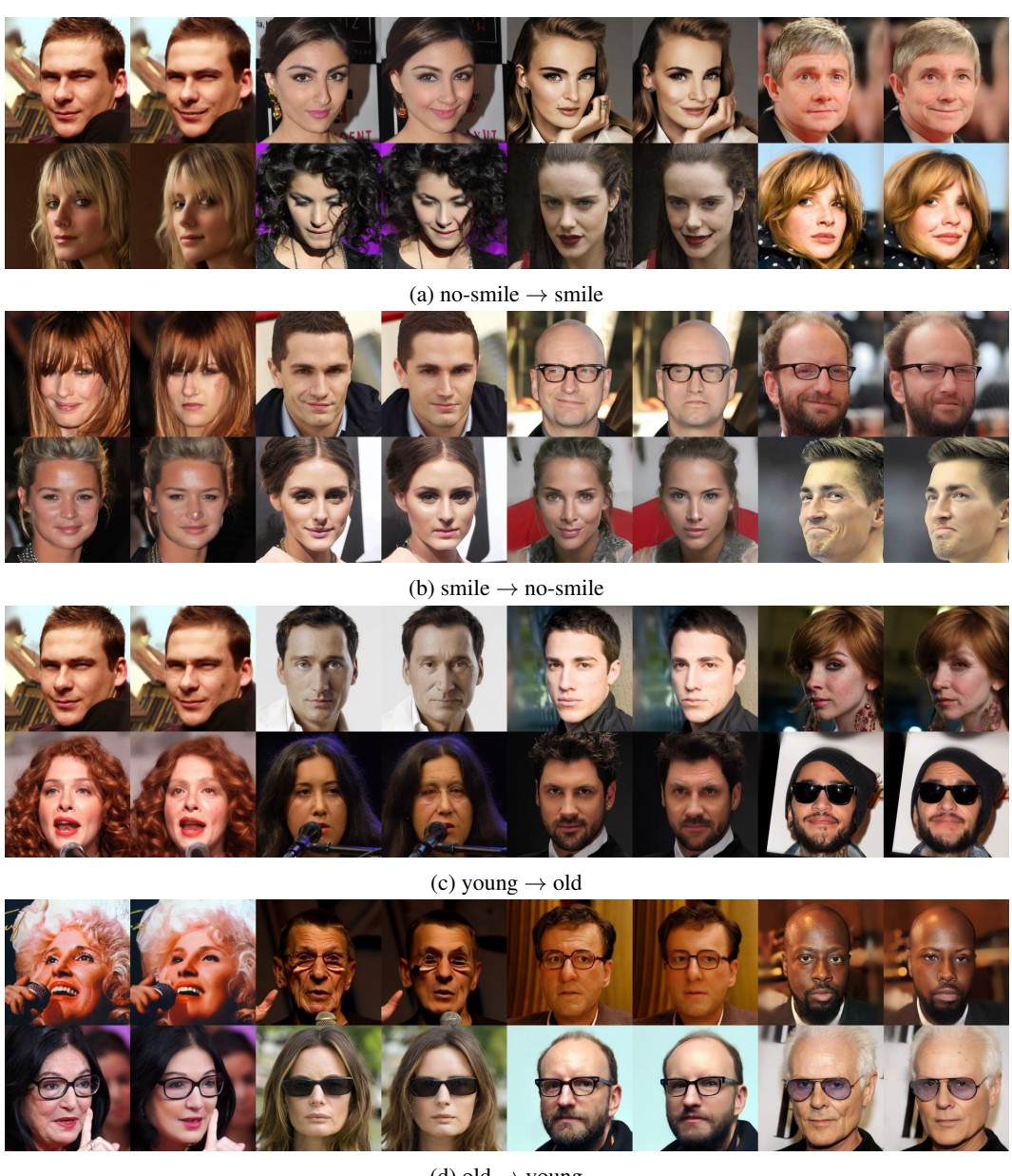

(a) no-smile → smile

(b) smile → no-smile

(c) young → old

(d) old → young

Figure 11: Additional qualitative results for LDCE-txt on CelebA HQ with DenseNet-121. Left: original image. Right: counterfactual image.

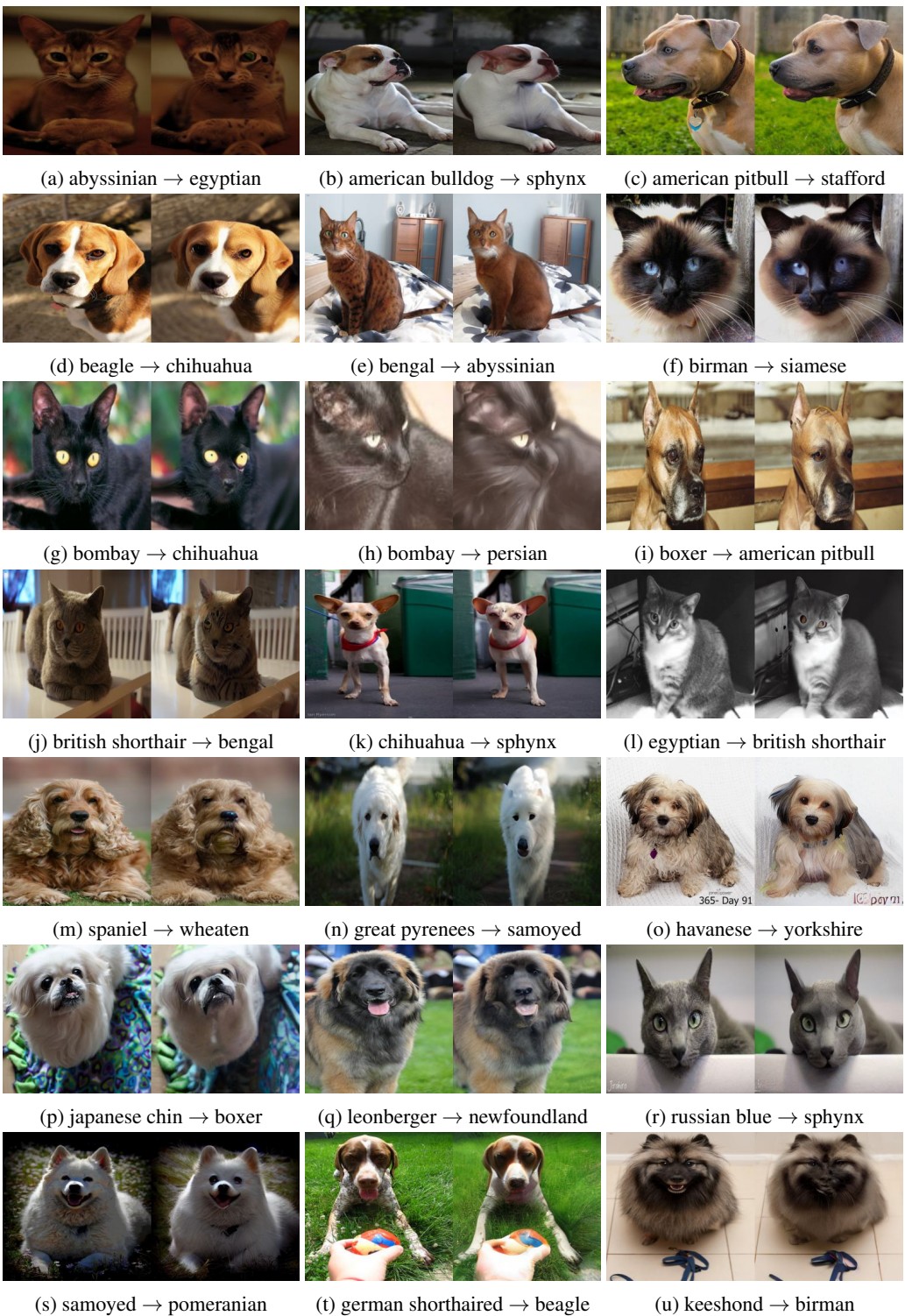

Figure 12: Additional qualitative results for LDCE-txt on Oxford Pets with OpenCLIP VIT-B/32. Left: original image. Right: counterfactual image.

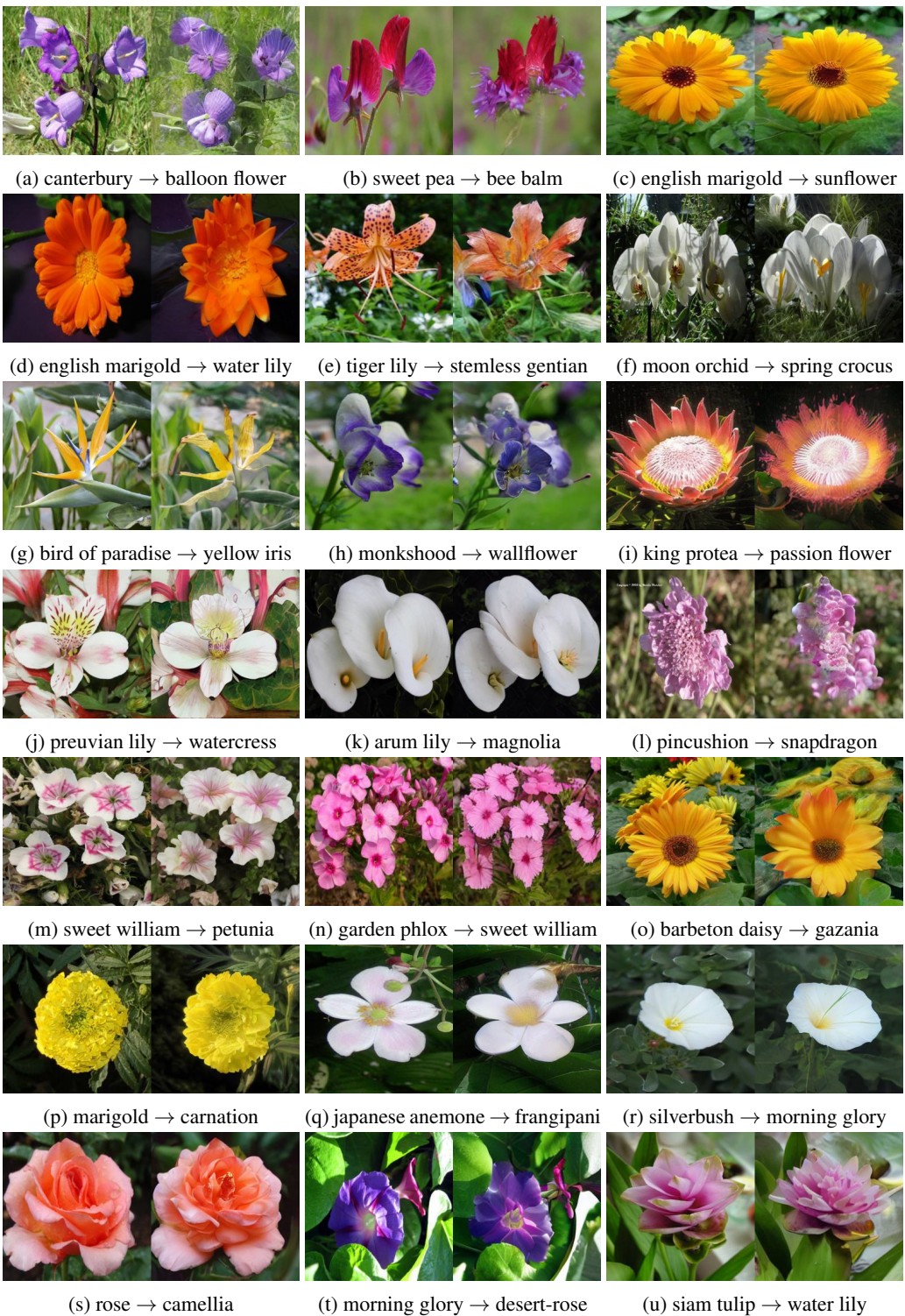

(a) canterbury → balloon flower     (b) sweet pea → bee balm     (c) english marigold → sunflower

(d) english marigold → water lily     (e) tiger lily → stemless gentian     (f) moon orchid → spring crocus

(g) bird of paradise → yellow iris     (h) monkshood → wallflower     (i) king protea → passion flower

(j) preuvian lily → watercress     (k) arum lily → magnolia     (l) pincushion → snapdragon

(m) sweet william → petunia     (n) garden phlox → sweet william     (o) barbeton daisy → gazania

(p) marigold → carnation     (q) japanese anemone → frangipani     (r) silverbush → morning glory

(s) rose → camellia     (t) morning glory → desert-rose     (u) siam tulip → water lily

Figure 13: Additional qualitative results for LDCE-txt on Oxford Flowers 102 with (frozen) DINO-VIT-S/8 with (trained) linear classifier. Left: original image. Right: counterfactual image.

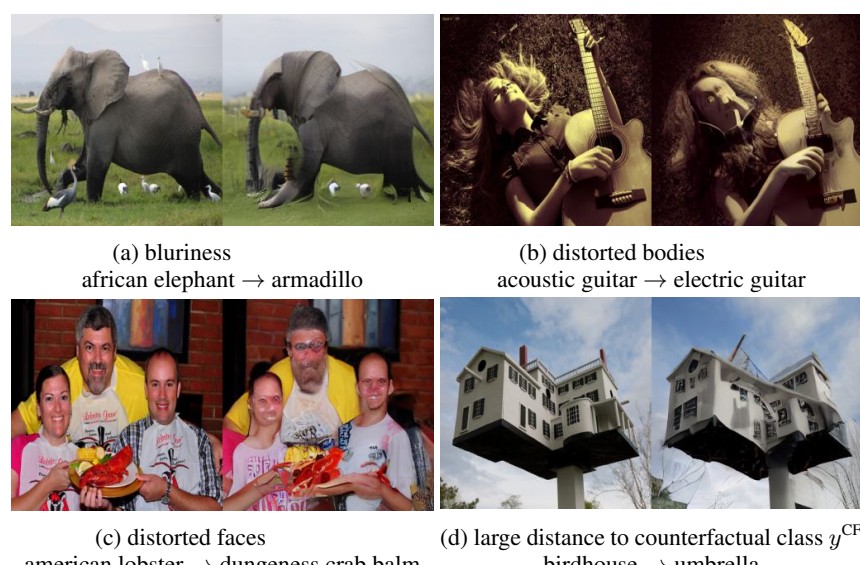

(a) bluriness
african elephant → armadillo

(b) distorted bodies
acoustic guitar → electric guitar

(c) distorted faces
american lobster → dungeness crab balm

(d) large distance to counterfactual class $y^{\text{CF}}$
birdhouse → umbrella

Figure 14: Failure modes of LDCE (i.e., LDCE-txt) on ImageNet (Deng et al., 2009) with ResNet-50 (He et al., 2016). Left: original image. Right: counterfactual image.

## K  FAILURE MODES

In this section, we aim to disclose some observed failure modes of LDCE (specifically LDCE-txt): (i) occasional blurry images (Figure 14(a)), (ii) distorted human bodies and faces (Figures 14(b) and 14(c)), and (iii) a large distance to the counterfactual target class causing difficulties in counterfactual generation (Figure 14(d)). Moreover, we note that these failure modes are further aggravated when multiple instances of the same class (Figure 14(c)) or multiple classes or objects are present in the image (Figure 14(a)).

As discussed in our limitations section (Section 5), we believe that the former cases (i & ii) can mostly be attributed to limitations in the foundation diffusion model, which can potentially be addressed through orthogonal advancements in generative modeling. On the other hand, the latter case (iii) could potentially be overcome by further hyperparameters tuning, e.g., increasing classifier strength $\lambda_c$ and decreasing the distance strength $\lambda_d$. However, it is important to note that such adjustments may lead to counterfactuals that are farther away from the original instance, thereby possibly violating the desired desiderata of closeness. Another approach would be to increase the number of diffusion steps $T$, but this would result in longer counterfactual generation times. Achieving a balance for these hyperparameters is highly dependent on the specific user requirements and the characteristics of the dataset.

