# OpenReview forum: "Latent Diffusion Counterfactual Explanations"
_ICLR.cc/2024/Conference — Submitted to ICLR 2024_

### Official Review · Reviewer_nXBG · 2023-10-29

**Soundness:** 3 good
**Presentation:** 3 good
**Contribution:** 3 good
**Rating:** 6
**Confidence:** 3

**Summary:**

This paper proposes a model-agnostic and computationally efficient diffusion model-based framework that can generate counterfactual explanations. Specifically, the model leverages the gradient of the classifier of a conditional diffusion model to filter out the gradient that is not semantically relavent or unimportant, which is natual and reasonable.

**Strengths:**

(1) The problem that the paper aims to solve is significant anf appealing.
(2) The proposed method that uses gradient of the classifier of a conditional diffusion model is straitforward and novel to me.
(3) They showed good performance in the experimental section, indicating the effectiveness of the method.

**Weaknesses:**

(1) It seems that the counterfactual explanation explained in the paper is very relavent to "semantic consistency" in [1]. I would suggest the author also discuss this paper.
(2) How will the angular threshold affect the model performance? My concern is that if we mask too much then the model might loose too much information to reconstruct the image.

[1] Li et al., Optimal Positive Generation via Latent Transformation for Contrastive Learning

**Questions:**

Please refer to my comments in "Weakness".

---

> ### Author Response · Authors · 2023-11-14
> **Re: Official Review of Submission4986 by Reviewer nXBG**
>
> We thank the reviewer for the constructive review. We are glad that the reviewer finds our method “straightforward and novel” and acknowledges that it “showed good performance”. Below we address the remaining concerns.
>
> ### Related work
>
> We are grateful that you brought this work [1] to our attention. We have included it in the updated manuscript. We would briefly mention the similarities as well as differences to our work for sake of completeness. Similar to our work, they also generate images from a latent space of a generative model that remain close to the original image. However, their focus is on obtaining positives for contrastive learning that maintain the semantics (i.e., class). In contrast, our work aims to understand which input features affect the classification of a classifier, thereby we also allow for (and promote) semantic changes.
>
> ### Impact of the angular threshold
>
> The angular threshold controls how much the gradients of the classifier are allowed to deviate from the implicit classifier without being considered adversarial. In the case of large masking (i.e., small angular threshold) this would often remove such gradients and replace them with some overwrite value (in our experiments zeros). Thus, we would have little to no guidance from the classifier. Consequently, our method would only receive guidance from the distance gradients and thereby reconstruct the original image. On the other hand, in case of a large angular threshold, our guidance mechanism will bypass adversarial changes that will flip the decision of the classifier without a semantic change in the counterfactual image.
>
> ---
>
> [1] Li, Yinqi, et al. "Optimal Positive Generation via Latent Transformation for Contrastive Learning." NeurIPS 2022.

---

### Official Review · Reviewer_LUsN · 2023-10-31

**Soundness:** 3 good
**Presentation:** 3 good
**Contribution:** 3 good
**Rating:** 6
**Confidence:** 2

**Summary:**

This paper introduces the Latent diffusion counterfactual explanation which uses latent diffusion models to expedite counterfactual generation and focus on the important semantic parts of the data. They propose a novel consensus guidance mechanism to filter out noisy,
adversarial gradients that are misaligned with the diffusion model’s implicit classifier.

**Strengths:**

The paper proposes a novel approach using class or text foundational diffusion models to generate counterfactual explanations that are both model and dataset-agnostic. The consensus guidance mechanism seems interesting and novel.

**Weaknesses:**

The paper presents a comprehensive study focused on images, but it appears that experiments on tabular datasets are missing. Tabular data is frequently encountered in various applications such as finance, healthcare, and retail, where counterfactual explanations are of significant interest (i.e., loan approval dataset).  Would your proposed method extend to these settings?

The paper indicated that their method expedites the counterfactual generation process in the main paper however they claim it is slow in the limitations section.

**Questions:**

Address the question last section.

---

> ### Author Response · Authors · 2023-11-14
> **Re: Official Review of Submission4986 by Reviewer LUsN**
>
> We thank the reviewer for the constructive review. We are glad that the reviewer finds our consensus guidance mechanism “interesting and novel”. Below we address the remaining concerns.
>
> ### Does our method extend to tabular data?
>
> We note that the focus of our paper lies in visual counterfactual explanations, similar to previous works, e.g., [1,2]. Thus, we hope that the reviewer understands that generating counterfactuals for tabular data would be out-of-scope for the present work but may be a valuable future direction.
>
> Despite this, we still would like to outline how our approach could be used for tabular data in future works. First, one would need to train a diffusion model, such as TabDDPM [3], on some tabular dataset, such as adult or churn2. Second, we would need to adjust the backward diffusion process, i.e., adopt the consensus guidance mechanism, and can then generate counterfactuals for tabular data.
>
> ### Speed-ups of our method
>
> Our method indeed significantly speeds-up counterfactual generation *compared to previous work* (see the penultimate paragraph in Section 4.2). However, the counterfactual generation process *cannot be considered fast overall*, as mentioned in our limitations section. Thus, it is a meaningful direction to further speed it up. One possible direction could be to distill the classifier behavior into the diffusion model. This effectively avoids the computationally-intensive computation of gradients at every iteration step and could significantly expedite counterfactual generation.
>
> Complementary to aforementioned, another benefit of our method is its seamless integration with diffusion models, ensuring that any advancement in speed in the recently very active research field of (text-conditional foundation) diffusion models, e.g., stable diffusion, would directly echo in speed-ups for our method. In a similar vein, our method benefits from improved generative capabilities of such diffusion models.
>
> ---
>
> [1] Goyal, Yash, et al. "Counterfactual visual explanations." ICML 2019.
>
> [2] Augustin, Maximilian, et al. "Diffusion visual counterfactual explanations." NeurIPS 2022.
>
> [3] Kotelnikov, Akim, et al. "Tabddpm: Modelling tabular data with diffusion models." ICML 2023.

---

> > ### Comment · Reviewer_LUsN · 2023-11-23
> >
> > Thank you for the clarification on the tabular dataset and the speed of your method. I will keep my score on the accept side.

---

### Official Review · Reviewer_wLyZ · 2023-11-01

**Soundness:** 2 fair
**Presentation:** 2 fair
**Contribution:** 1 poor
**Rating:** 3
**Confidence:** 3

**Summary:**

Counterfactual explanation aims to find the realistic input data with the smallest semantically meaningful change that results in the counterfactual output.
To obtain such fake-but-counterfactually-realistic data, the authors utilized latent diffusion models and arbitrary classifier can be jointly adopted.
The authors argue that (1) the diffusion process in the latent space allows the proposed LDCE to focus on the important semantics of the data rather than the unimportant details; and (2) the proposed threshold-based consensus guidance mechanism against the implicit classifier can filter out meaningless adversarial gradients so that it enables capturing meaningful gradients for the counterfactual sample.

**Strengths:**

Strength

- The proposed method is very simple and easy to apply.
- The paper is easy to follow in general.
- The proposed method can be utilized to any LDMs.
- The implementation is properly provided for the reproducibility.

**Weaknesses:**

Weakness

- The paper should be written in more formal way.
- There lacks explanation on Algorithm 1 in the main body of the paper, which makes it hard to understand the technical connection between the proposed method and algorithm.
- Some generated counterfactual examples seem to be unrealistic. (Possibly, the threshold-based gradient filter cannot properly filter out the adversarial gradients?)
- The authors argued that the proposed threshold-based consensus guidance mechanism filters out the meaningless adversarial gradients, but when it comes to Figure 4, it seems that those the consensus guidance mechanism cannot filter out those gradients. For example, in Figure 4(f), intervening "bulldog" to "beagle" should not affect the grass texture since it should not be sensitive to the breed of dogs. It seems that those factors are not perfectly disentangled in the latent or noise space.
- Definitely, the proposed method is quantitatively outperformed in the CelebA case.

**Questions:**

Question

Could you provide more detailed explanation on Algorithm 1? How does it derived from Equation 10 and 11?

---

> ### Author Response · Authors · 2023-11-14
> **Re: Official Review of Submission4986 by Reviewer wLyZ**
>
> We thank the reviewer for taking the time to review our paper and provide constructive criticisms. We are happy to see that the reviewer finds that our method is “very simple and easy to apply” and that the paper is “easy to follow” overall. Below we address remaining concerns or questions of the reviewer.
>
> While appreciating the valuable feedback by the reviewer, we feel that some of the scores, such as the contribution score of 1, may not be entirely reflected by the written feedback. For the sake of the discussion of our paper, we kindly ask for further clarification from the reviewer. We believe this will greatly contribute to the discussion of our paper.
>
> ### Request for more formal writing
>
> We would appreciate it if the reviewer could point out parts of the submission that should be improved w.r.t. their formality.
>
> ### Added explanation to Algorithm 1
>
> As suggested by the reviewer, we added a brief explanation of Algorithm 1 in the main body.
>
> ### Some unrealistic counterfactual examples (e.g., Fig. 4(f))
>
> We agree that the generated counterfactual examples, e.g., Fig. 4(f), are not perfect. However, note that they are better than the ones from previous works, e.g., Fig. 3(h), where the problem of smoothing of high-frequency details is also prevalent. The deficiency in preserving high-frequency details in LDCE also stems from the application of the counterfactual generation within the lower-dimensional latent space. Here, the autoencoder’s decoder just fills in the high-frequency details. Thus, a better decoder, such as OpenAI’s recent consistency decoder [1], may do a better job filling in these fine details. Besides aforementioned, note that, as can be observed in Fig. 3(h), that changes in grass texture may also indicate a model bias, which may be desirable in the context of counterfactual generation.
>
> ### Discussion on Celeb-A HQ results
>
> The experiments on Celeb-A HQ demonstrate that our method, LDCE, is competitive to methods that are *specifically tailored* for this dataset, i.e., they use generative models pretrained on Celeb-A HQ. In fact, our method typically achieves the first or second rank on most metrics (Table 6) and is only inferior to ACE on Celeb-A HQ.
>
> However, we find that LDCE is clearly superior to ACE on the more challenging ImageNet dataset (Table 2) at the same time. Arguably, “ImageNet is extremely complex and the classifier needs multiple factors for the decision-making process” [1, p.5] as argued by the authors of ACE. Note that this comparison is an apples-to-apples comparison of the methods, as LDCE-cls also uses a diffusion model that has been trained on ImageNet. It is worth noting that LDCE-txt, which uses stable diffusion as a diffusion model, is also superior to ACE.
>
> Henceforth, we posit that the diminished performance observed on the domain-specific facial data from CelebA HQ can be attributed to the characteristics of the underlying diffusion model used in a specific counterfactual method. In fact, stable diffusion is renowned for its challenges in accurately generating faces and individuals [2]; a point openly acknowledged in both our limitations section and Appendix K. Nonetheless, the inherent merit of employing a non-domain-specific generative model - which has not been done before for counterfactual generation with generative models to the best of our knowledge - lies in its versatility, rendering it applicable across diverse datasets (as demonstrated on CelebA HQ). This flexibility becomes particularly valuable in real-world scenarios, where constraints on data accessibility are prevalent or pose significant challenges.
>
> Though, if one has access to domain data, it is definitely beneficial to utilize it. To adapt our method, one could use low-rank adaptation that finetunes an existing stable diffusion model without the need of extensive computational resources. We expect that this will improve performance also on domain-specific data distributions, such as Celeb-A HQ.
>
> ---
>
> [1] https://github.com/openai/consistencydecoder
>
> [2] Jeanneret, Guillaume, Loïc Simon, and Frédéric Jurie. "Adversarial Counterfactual Visual Explanations." CVPR 2023.
>
> [3] https://huggingface.co/stabilityai/stable-diffusion-2

---

### Author Response · Authors · 2023-11-22
**End of the discussion period is approaching**

Dear reviewers,

As the end of the discussion period approaches, we would truly appreciate it if you would engage with our responses. If you still have any concerns or questions, we would be happy to hear from you. Thank you in advance!

---

### Meta-Review · Area_Chair_J5zP · 2023-12-06

**Metareview:**

This paper introduces Latent Diffusion Counterfactual Explanations (LDCE) to address challenges in generating counterfactual explanations for opaque black-box models. Leveraging latent diffusion models, LDCE efficiently generates meaningful counterfactuals without relying on auxiliary adversarially robust models or computationally intensive guidance schemes. The proposed consensus guidance mechanism filters out noisy, adversarial gradients, enhancing the robustness of the explanations.

The paper got controversial comments from the reviewers. While most of the reviewers recognize the technical novelty and effectiveness of this paper, one reviewer has some concern on the technical soundness of the proposed method. In particular, the motivation and the reason why the proposed consensus mechanism works is not clearly explained. In addition, in the experimental results, the method with no consensus still performs better in terms of three out of five metrics. I would suggest the authors further improve the paper and consider future submissions.

**Justification For Why Not Higher Score:**

The reason why the proposed method is not clearly explained. The experimental results lack consistency on different evaluation metrics.

**Justification For Why Not Lower Score:**

n/a

---

### Decision · Program_Chairs · 2024-01-16

Reject